# The high-dimensional space of human diseases built from diagnosis records and mapped to genetic loci

Gengjie Jia [1,20,21] ✉, Yu Li[2,3,4,20], Xue Zhong[5], Kanix Wang [6,7],
Milton Pividori [6,8], Rabab Alomairy [9,10], Aniello Esposito[11], Hatem Ltaief[9],
Chikashi Terao [12,13,14], Masato Akiyama[12,15], Koichi Matsuda[16], David E. Keyes[9],
Hae Kyung Im[6], Takashi Gojobori[2,17], Yoichiro Kamatani [12,16], Michiaki Kubo[12],
Nancy J. Cox[5], James Evans [18], Xin Gao [2,3,21] ✉ & Andrey Rzhetsky [6,19,21] ✉

Human diseases are traditionally studied as singular, independent entities, limiting researchers' capacity to view human illnesses as dependent states in a complex, homeostatic system. Here, using time-stamped clinical records of over 151 million unique Americans, we construct a disease representation as points in a continuous, high-dimensional space, where diseases with similar etiology and manifestations lie near one another. We use the UK Biobank cohort, with half a million participants, to perform a genome-wide association study of newly defined human quantitative traits reflecting individuals' health states, corresponding to patient positions in our disease space. We discover 116 genetic associations involving 108 genetic loci and then use ten disease constellations resulting from clustering analysis of diseases in the embedding space, as well as 30 common diseases, to demonstrate that these genetic associations can be used to robustly predict various morbidities.

It is convenient to consider diseases as distinct, well-defined, objective entities, a pattern reinforced by disease taxonomies that divide diseases by topographical, anatomical and cultural similarities. Disease definitions often cover overlapping collections of symptoms; distinct diseases can be etiologically linked, result in the same downstream health conditions and co-occur in the same patient. One disease may change the clinical symptoms, prognoses and characteristics of another. Academic medicine recognizes complex associations between diseases, but scientists have previously considered these between pairs of diseases and only recently across the disease system as a whole[1–3]. There are prior studies aiming to infer disease–disease relationships based on genetic overlaps and comorbidity[4], protein–protein interactions[5], shared metabolism[6] and multiple types of input data at the same time[7]. The increasing availability of large-scale electronic health records (EHRs) has enabled researchers to identify disease–disease

relationships on a large scale. There is a set of studies using topological methods for disease analysis, focusing on one or a few specific diseases[8]. These analyses used EHRs for topological inference, sometimes incorporating the temporal order of diseases in a patient's history, producing topological disease networks[9]. Yet another group of studies generates disease networks by computing pairwise disease–disease correlations or relative risk scores[10,11].

In this study, we set out to develop and test a representation of the complete disorder spectrum as points in a continuous, metric, high-dimensional disease space, which we then linked to the underlying genetic space to enable clinical prediction and etiological discovery. Word embedding procedure was invented to capture the semantic similarities of words and phrases in a natural language[12,13]. It transforms discrete high-dimensional one-hot representation of millions of words into a real-valued, low-dimensional space. The typical dimensionality

of a word embedding is in the low hundreds. Each word is a point in this space, and semantically close words are located close to each other in the embedding space. In this study, we applied a word embedding technique to map diseases into a 20-dimensional embedding space, enabling various downstream analyses[14]. Here, we analyze large-scale clinical data, describing human health states as points in a continuous disease space (see Supplementary Fig. 1 for the overall workflow of this study).

## Results

### Disease embedding space development

To compute disease space, we considered chronologically ordered patient health histories as context. We then used a shallow neural network model[12,13] to predict a randomly deleted diagnosis from its disease context, treating diseases as words in a text. As input data, we used the Merative MarketScan dataset[15–17] (Methods), a massive clinical dataset representing more than 151 million patient health histories, where diagnoses were restricted to 547 broadly-defined diseases, such as 'asthma' or 'depression.' Each disease label was typically represented by multiple International Classification of Diseases codes[18], with no overlapping codes among distinct diseases. We then obtained a 20-dimensional continuous-space embedding of these 547 disease categories, and applied the principal component analysis to rotate the disease vectors in such a way that dimension 1 corresponds to the highest variance principal component (PC), all the way to dimension 20, which corresponds to the twentieth PC (Methods).

The word2vec model[12,13] trained by removing a disease from a sliding window of diseases and learning to predict the missing disease from its context. Consequently, each disease is mapped to a 20-dimensional point, and thus 547 diseases form clusters of points in the disease space, with dimension numbers treated as coordinates. Because it is difficult to visualize an object in a 20-dimensional disease space directly, we proposed multiple projections of it to facilitate an understanding of its properties. As one type of projection, Fig. 1a shows a three-dimensional projection of the disease space through the t-Distributed Stochastic Neighbor Embedding (t-SNE) algorithm[19], with an emphasis on neuropsychiatric diseases (three shades of red), infections (green) and cancers (yellow). We also retained all 20 dimensions, picturing three diseases at a time, for selected neuropsychiatric and infectious diseases (Fig. 1b,c). As expected, neuropsychiatric conditions are, generally, much closer to each other in the disease space than a neuropsychiatric disease is to an infectious disease.

### Interpretation of the disease space

To glimpse the properties of our disease embedding space, we represented disease proximity in terms of angles between disease-specific vectors using cosine similarity. As shown in Fig. 2a, a collection of degenerative diseases of the central nervous system (CNS) have fasciitis (a typical athlete's injury) as their most distant counterpart in the disease space. The most distant disease for migraine is unspecified diabetes mellitus, and for obsessive–compulsive disorder, the 'antipode' disease is acute renal failure. These antipode diseases are very unlikely to coexist within the same body or lead to or closely follow one another. Figure 2b depicts the closest counterparts for the same set of diseases shown in Fig. 2a. Thus, Fig. 2b can serve as a control.

The disease embedding space captures higher order, etiological relationships between diseases, enabling the computational discovery of hidden disease analogies. For example, because each disease in our representation corresponds to a 20-dimensional vector, we can discover the following approximate relations between disease vectors using vector algebra: (chondrocalcinosis) + (connective tissue infection) ≈ (septic arthritis), and (abnormal spine curvature) – (congenital spine anomaly) + (gout-related crystal arthropathies) ≈ (chondrocalcinosis). These established disease analogies can be informative in understanding the combinatoric properties of complex diseases.

Considering one dimension of our disease space at a time, we ordered all 547 diseases according to their positions in this dimension, from the most negative value to the most positive. For example, Fig. 2c shows how dimension 8 orders diseases. Along this dimension, male genital congenital anomaly is at the most negative end, while the largest value is associated with complex regional pain syndrome. In the same way, we show the disease orders for all the other 19 dimensions in Extended Data Fig. 1. To further interpret the meaning of our disease space dimensions, we assigned each dimension a label after identifying the disease category pairs that are statistically significantly separate along this dimension. We grouped the 547 unique diseases into 21 general disease categories, and for each pair of disease groups, we performed the Wilcoxon rank sum test to judge how significantly these groups separate from each other[20–22] (see Methods for technical details and Supplementary Data 1 for test results for this analysis). For example, dimension 8 received the label 'ophthalmological (−) to CNS (+)' and dimension 2 was labeled 'neuropsychiatric (−) to infectious (+)' (Supplementary Fig. 2).

We further implemented a singular value decomposition (SVD) approach[23,24] to identify ten maximally distinct disease constellations that anchor the disease embedding space. Individual disease projections onto these ten constellations are akin to resolved 'disease clusters' (Methods; these clusters can be similarly found using hierarchical clustering methods, as reported in Supplementary Data 2). In Fig. 3, we show how diseases are partitioned into ten disease constellations (Supplementary Data 3) and label the diseases found most similar to each respective constellation. For example, CNS diseases, such as dopamine-responsive dystonia and Parkinson's disease, have the highest cosine similarities with disease constellation 1.

### Human health states and corresponding genetic associations

Just as weight and height measurements can map each patient to a point on a two-dimensional weight–height surface, each patient's specific health history places the patient at a unique point in the disease embedding space. By analogy with height and weight, each of our disease space's 20 dimensions can be treated as a continuous measurement, or trait. If a patient has multiple diseases, we represent her health state as a weighted mean of her disease coordinates (Methods). By interpreting disease embedding space dimensions as continuous traits, we can genetically map them using standard methodology, such as family pedigree analysis and genome-wide association studies (GWASs).

Figure 4a shows shared-parent environment contributions, and family- and GWAS-based heritability estimates for all dimensions in the disease embedding space. We obtained these estimates from an analysis of 128,989 nuclear families with the fullest medical history recorded in the MarketScan, which included children aged at least 16 and at least 15 years younger than parents. The resulting 481,657 individuals had been enrolled in the database for an average of 6.5 years.

We additionally obtained GWAS heritability estimates using the BioBank Japan (BBJ) cohort, an East Asian cohort of patients with documented common diseases and genotypes[25–27]. GWAS-based heritability estimates appear much smaller than family-based heritability, as is typically the case in such comparisons. Our analysis suggests that among all dimensions, dimension 19, 'immune (−) to neuropsychiatric (+)' has both the largest family heritability and the largest contribution of parental environment, $e^2$. Dimension 15, 'metabolic (−) to infectious (+)', has the lowest values of these family-based estimates. As shown in Fig. 4b, 20 continuous traits possess complex patterns of pairwise genetic correlations, estimated using both the family data (the upper right triangular matrix) and the BBJ's GWAS data (the lower left triangular matrix). Estimates are distinct across the two approaches, but are significantly positively correlated ($r = 0.224$, with a 95% credible interval (0.085, 0.355), $P = 1.85 \times 10^{-3}$). The individual heritability of a single dimension does not rise above 0.2 and, due to high genetic

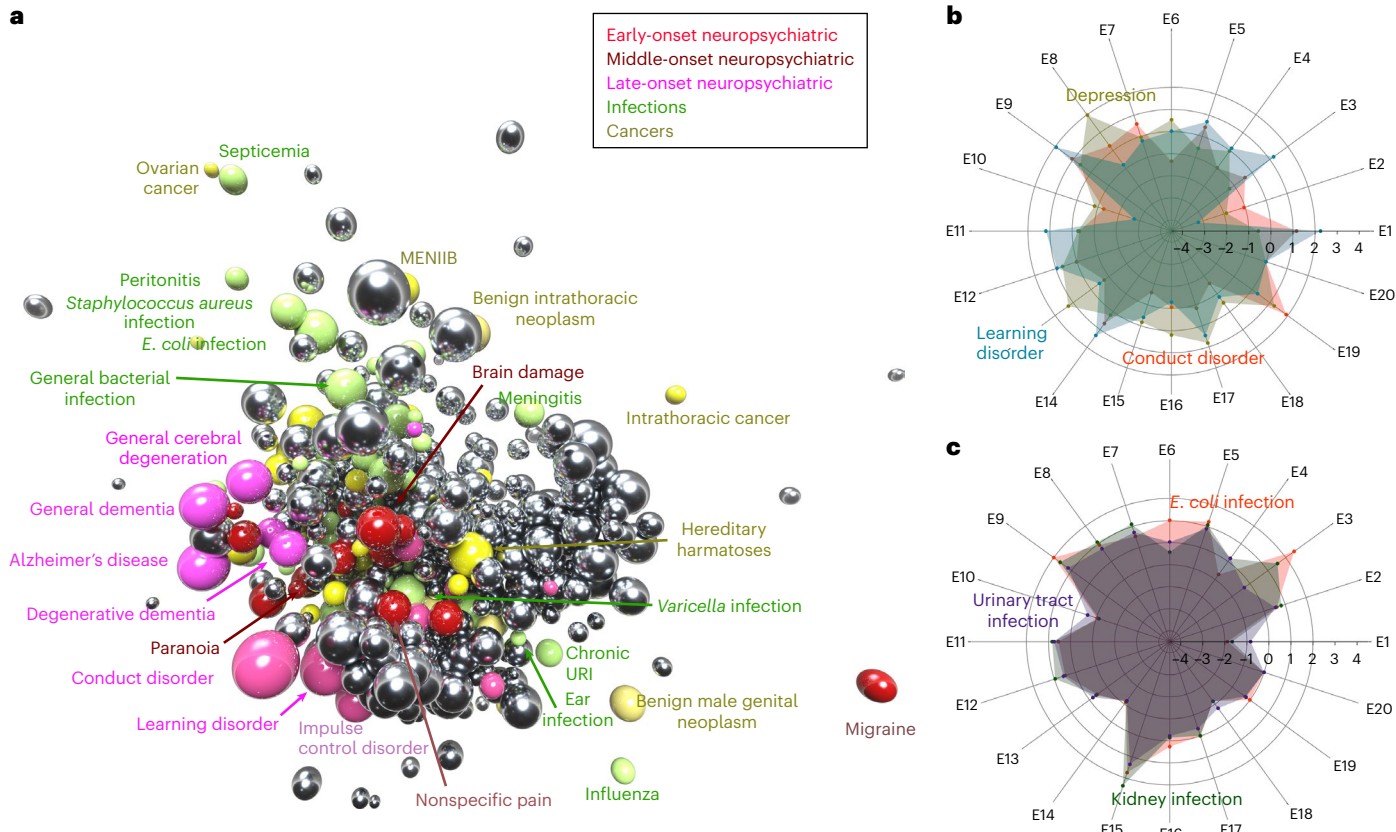

**Fig. 1 | Disease embedding space representations. a**, Three-dimensional mapping. To enable visualization of the disease embedding space, while preserving the neighborhood structure of the original space, we used the *t*-SNE algorithm to project the entire 20-dimensional disease space onto a three-dimensional, metric space. In this plot, individual spheres correspond to diseases, with sphere volume indicating disease prevalence, and sphere color encoding disease class, as stated in the key. We show neuropsychiatric diseases stratified by the age of onset (early, middle or late) in three shades of red, infectious diseases in green and cancers in yellow. **b**,**c**, Radar charts to show points that correspond to individual disease coordinates once embedded in the disease space. Each of the disease's 20 coordinates maps to a point on the corresponding radial axis (labeled E1 to E20). A patch connecting all 20 coordinates for a single disease forms a polygon specific to this disease. Diseases belonging to the same class produce more similar radar plots, and are more similar among themselves than those across distinct disease classes. **b**, Plots representing three neuropsychiatric diseases (depression, learning disorder and conduct disorder) are more similar to each other than any of the neuropsychiatric plots that represent infectious diseases shown in **c**. **c**, Radar plot for infectious diseases, including *Escherichia coli infection*, urinary tract infection and kidney infection.

---

correlation between dimensions, the joint co-heritability of all 20 dimensions is 0.187.

Next, we mapped the genetic associations for the 20 dimensions, using the largest genetic cohort, the UK Biobank (UKB) data[28] (Methods). Figure 4c shows the results: the first ten dimensions' GWASs are shown at the top panel, and the remaining GWASs are at the bottom. The plot's *y*-axis shows the negative base-ten logarithm of the corresponding association's *P*-value, with dotted lines indicating the genome-wide significance level. We color-coded each dimension and annotated significant GWAS associations with their nearest gene names in dimension-specific colors.

Our 20 'quantitative traits' yielded 116 association signals that reach the genome-wide significance level (after a Bonferroni correction for multiple testing). These associations involve 108 unique genetic loci, 40 of which have never been reported in any historical GWASs (Supplementary Data 4 and 5).

For example, dimension 8 associates with six loci near genes *BACH2*, *GTF3AP1*, *STAT6*, *EMSY* and *SMAD3*. These genes are transcription factors involved in immune response, including interleukin 4-mediated signaling, class switch of immunoglobulins, T-cell function and B-cell maturation. Therefore, these genes' biology may point to molecular mechanisms underlying diseases like migraine and inflammatory bowel syndrome (IBS). Analysis of the GWAS Catalog suggested

that dimension 8 relates to processes underlying changes of eosinophil counts and changes in the proportion of neutrophils in granulocytes, and effect allergic diseases, such as asthma and hay fever.

Immunity, such as interleukin 1 family signaling, is represented in dimensions 1 and 10. Dimension 1 separates infectious and developmental diseases as two extremes, while dimension 10 separates infectious and hematologic diseases. It is likely that the immune system is involved to some extent in every complex disease, and these two dimensions capture interactions of weakened or over-active immunity with other biological systems (see Supplementary Data 6 for the enrichment results specific to each of the 20 dimensions, and Methods for technical details).

We validated these 116 associations in independent, albeit smaller, cohorts from two additional countries, Japan and the United States of America (USA) (the BBJ[25–27], and BioVU from Vanderbilt University's Medical Center[29], respectively; Methods and Table 1). 

In the BBJ cohort, we found 87 out of 100 associations to be in the same effect direction (a positive or negative sign of regression parameter) as discovered in the UKB cohort (the other 16 of the 116 discovered associations involved 15 unique loci not genotyped or imputed in the BBJ; we thus excluded them for the replication attempts here). In the BioVU cohort, we found 87 of 116 associations in the same effect direction as discovered in the UKB cohort. There were 67 associations with

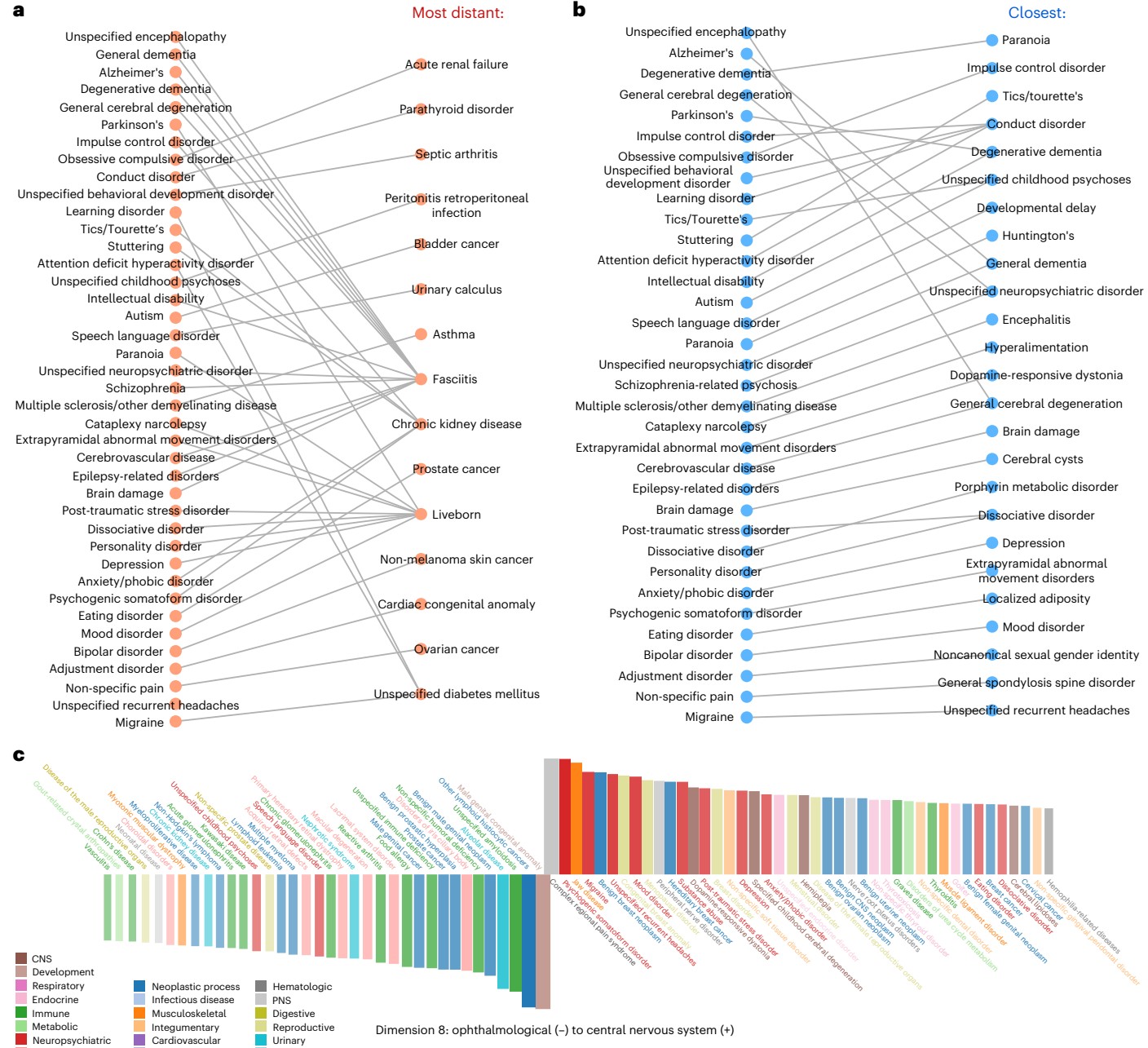

**Fig. 2 | Disease similarities and embedding-dimension-specific disease orderings. a**, Most-distant disease pairs. Because we can quantify the similarity between any disease pair represented as 20-dimensional points, we can also identify an antipode (the most distant disease) for each disease. The left column shows antipodes for neuropsychiatric, peripheral and CNS diseases. The right column shows the most distant disease for each disease in the left column (corresponding diseases are connected with an edge). For example, the most distant disease from unspecified encephalopathy is fasciitis. **b**, The closest disease neighbors for the same set of diseases. For instance, general cerebral degeneration is closest to unspecified encephalopathy. **c**, Diseases ordered by their positions along a dimension. We rank diseases in every dimension of

the disease embedding space according to the value of their corresponding coordinate. This chart shows dimension 8. We plot positive and negative values above and below the horizontal axis, respectively. We plot the largest absolute values at the middle and other, descending values from the middle to the right for positive values, and from the middle to the left for negative ones. The bars are color-coded by disease category. A group of neuropsychiatric, peripheral and CNS diseases, such as psychogenic somatoform disorder, complex regional pain syndrome and dopamine-responsive dystonia, are clustered on the right side of the plot (they have large positive values), while a group of cancers, such as benign male genital neoplasm and other lymphoid histiocytic cancers, cluster in proximity on the left side (they have large negative values).

the same effect directions in all three cohorts (Supplementary Fig. 3). After multiple testing corrections, 41 and 15 associations for BBJ and BioVU, respectively, remained significant at a false discovery rate (FDR) of 0.05 (ref. 30), in which eight significant associations overlapped. The loci involved in these eight, three-way overlapping signals among the

three cohorts are close to genes *LPA*, *CDKN2B-AS1*, *TCF7L2*, *HLA-B* and *HLA-DQA1* (Supplementary Data 4).

We expected that both clinical measurements and environmental factors would associate with specific areas or dimensions of the disease space. To test this conjecture, we analyzed 140 various

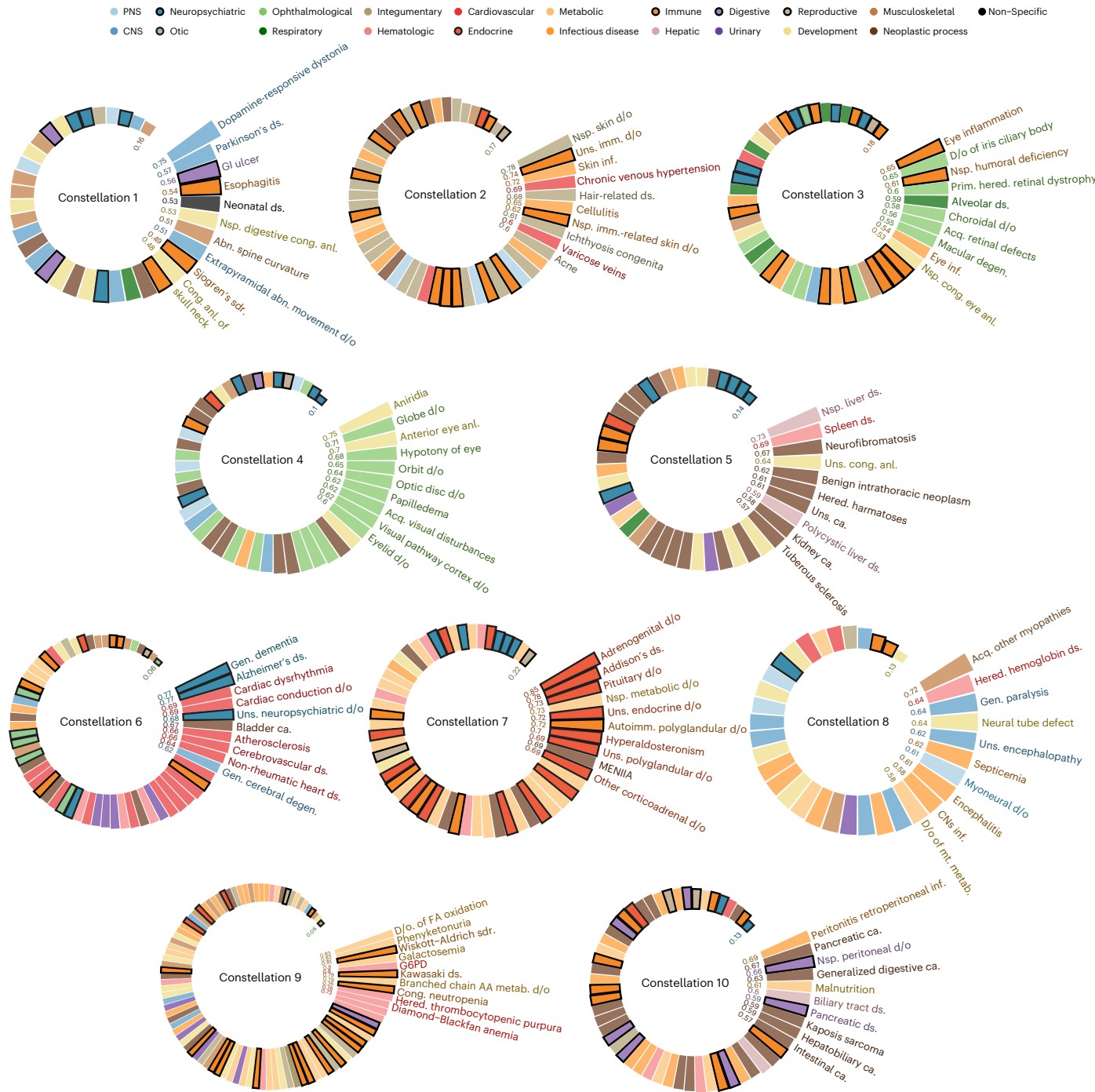

**Fig. 3 | Ten disease constellations discovered using the 20-dimensional disease embeddings.** First, we assigned all the diseases to their most similar disease constellations as shown in these spiral bar charts. Cosine similarity values between diseases and constellations are plotted as radial bars (values are shown inside the cycles) colored by the functional categories to which the disease belongs. Only the top ten most similar diseases are annotated here for each constellation (the complete disease list can be found in Supplementary Data 3).

AA, amino acid; abn., abnormal; acq., acquired; anl., anomaly; ca., cancer; cong., congenital; degen., degeneration; ds., disease; d/o, disorder(s); FA, fatty acid; gen., general; GI, gastrointestinal; hered., hereditary; imm., immune; inf., infection; MENIIA, multiple endocrine neoplasia type IIA; metab., metabolism; mt., mitochondrial; nsp., non-specific; PNS, peripheral nervous system; prim., primary; sdr., syndrome; uns., unspecified.

individual-specific features recorded in the UKB dataset (Methods). Following our discussion of similarities among migraine, IBS and eye inflammation, we highlight here the features associated with dimension 8. Counts of eosinophils and leukocytes were in strong negative associations with individuals' embedding values on dimension 8. We show the complete results in Supplementary Figs. 4–13 and Supplementary Data 7.

(Supplementary Fig. 4). Similarly, levels of creatinine and glycated hemoglobin HbA1c in blood (Supplementary Fig. 5) and of microalbumin in urine (Supplementary Fig. 6) are also negatively associated with dimension 8. We show the complete results in Supplementary Figs. 4–13 and Supplementary Data 7.

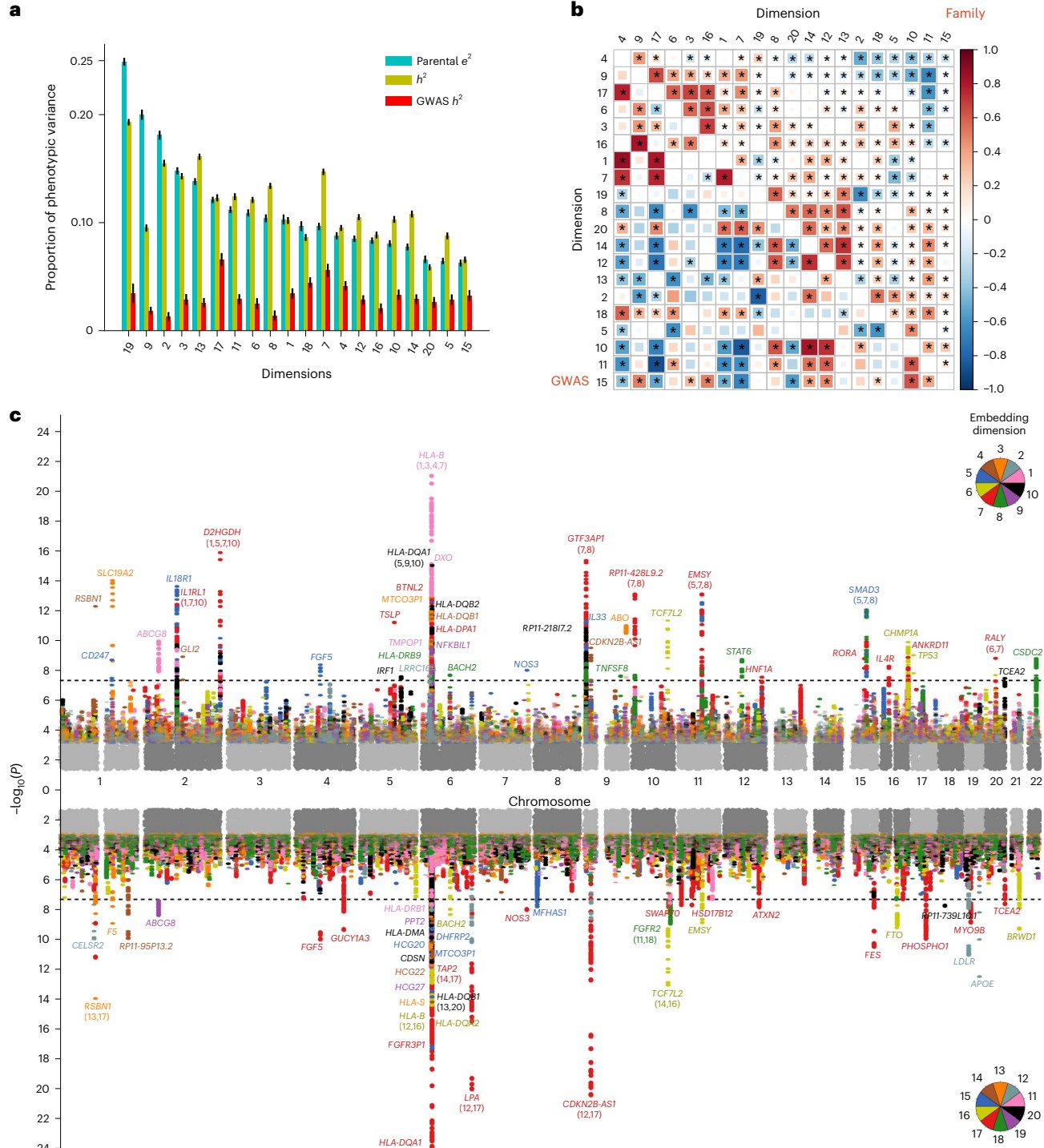

**Fig. 4 | Heritabilities, genetic correlations and genome-wide associations.**
**a**, Estimates of proportion of variance explained by genetic and environmental
factors. By viewing each of the 20 dimensions as a quantitative trait, we
computed three estimates for each dimension: family-based heritability ($h^2$),
shared-parental environmental factors ($e^2$) and GWAS-based $h^2$. $h^2$ and $e^2$ are
the estimates of the proportion of the total phenotypic variance explained
by additive genetic variations and by the environment shared by parents in a
nuclear family, respectively, while GWAS $h^2$ is an SNP-marker-based counterpart
of family-based $h^2$. Data are presented as mean estimates +/− standard errors.
We obtained the family-based estimates using the US MarketScan database, and
the GWAS estimates using the BBJ cohort. We ordered the dimensions on the
figure by decreasing $e^2$ values. **b**, Genetic correlation estimates. We estimated
genetic correlations using both the US MarketScan's family data (the upper right
triangular matrix) and the BBJ's GWAS data (the lower left triangular matrix). The

estimates are colored by sign and magnitude. The size of the colored squares in
cells indicates statistical significance, and asterisks indicate pairwise correlations
remaining significant at an FDR of 1%. **c**, Genome-wide associations with the 20
embedded disease dimensions. Here, we show a Miami plot for all 20 disease space
dimensions, where the results of the embedded dimensions 1 to 10 are overlaid
in the top panel and dimensions 11 to 20 are represented in the bottom panel.
Each point corresponds to a SNP and is color-coded by its associated embedded
dimension (see pie charts). The $y$-axis shows $-\log_{10}(P)$ and the $x$-axis represents
the chromosomal location of a given SNP. A dark grey, dashed line indicates the
genome-wide significance threshold ($P = 5 \times 10^{-8}$). In addition, we annotated
genome-wide significant loci by their nearest genes, and each embedded
dimension's serial number is written in parentheses under the gene name when a
gene is found in the results of multiple embedded dimensions. More details about
these annotated associations can be found in Supplementary Data 4.

**Table 1 | Subject characteristics and sample sizes**

|  | MarketScan claims | UK Biobank | BioBank Japan | BioVU |
|---|---|---|---|---|
| Description | Insurance claims in USA | National health database in UK | Patient-based registry in Japan | Patient-based registry of Vanderbilt University Medical Center |
| Males | 46.5% | 45.7% | 54.1% | 50.3% |
| Median age (years)[a] | 35 (17–51) | 59 (51–64) | 65 (55–73) | 61 (51–71) |
| Sample size | 122,740,623 | 306,629 | 166,612 | 16,545 |
| Used data types | Diagnoses | Diagnoses and genotypes | | |
| Usage | To develop disease embeddings | To discover genetic associations | To replicate genetic associations | |

Four large datasets, including the Merative MarketScan, UK Biobank, BioBank Japan and BioVU data were used in this study. We summarize their sample characteristics here. [a]Values in parentheses are interquartile ranges.

### The utility of polygenic prediction models

We attempted to use our newly found genetic associations in the disease embedding space to predict patients' disease predisposition. We started with the five most common diseases: asthma, allergic rhinitis, depression, general hypertension and osteoarthritis. Baseline data from historical analyses were available for three of these diseases, represented as Nagelkerke $R^2$ values, the proportion of the total outcome variability (the presence or absence of a disease) as explained by the model, where higher $R^2$ values indicate better models with greater explanatory power.

To test for genetics-based disease forecasting, we implemented three models. Our first model was a generalized linear model (GLM) using the known genetic associations for each disease reported in the GWAS Catalog[31]. The second model was almost identical to the first but used the same 116 SNP associations identified in our own analysis for every disease. The third model also relied on the set of 116 associations, the same for every disease, but instead GLM was replaced with the gradient boosting classification model (GBM)[32].

For each disease, we sampled standardized sets of 5,000 cases (individuals diagnosed with a particular disease) and 5,000 controls (this specific disease absent) for model training, and another 2,500 cases and 2,500 controls to test model performance. We repeated this model training and testing process 100 times for confidence interval estimation (Methods). In Supplementary Table 1, we compare each analysis with published Nagelkerke $R^2$ values; the third model performed the best, increasing $R^2$ from 0.025 (published)[33] to 0.06 for asthma, from 0.011 (published)[34,35] to 0.07 for depression, and from 0.035 (published)[36] to 0.14 for general hypertension. Thus, our data propose unanimous, massive gain in genetic variation's explanatory power. The prediction accuracy of discriminating cases from controls in the testing datasets based on genetic variation alone was under 63% for all diseases—and as low as 54% for asthma. We also reported the positive predictive value (PPV) and negative predictive value (NPV), which showed comparable results to the prediction accuracy value.

In addition to the five most-prevalent diseases, we checked the performance of our models for all the other diseases that had at least 15,000 case counts in the UK Biobank data, reserving each time at 5,000 cases for model training and at least 2,500 cases for model testing, that is, 25 diseases in total (Supplementary Table 2). Our disease-space-based models still provided a consistent gain in Nagelkerke $R^2$, compared with the conventional models built on disease-specific markers. Out of the 30 diseases we report here, there were 12 diseases for which the model prediction accuracies were greater than 60%.

Furthermore, we sought to predict whether a patient is likely to reach one of the ten disease constellations derived earlier (Methods). Table 2 summarizes the performance of our GLM and GBM models. Nagelkerke $R^2$ values are significant for all ten of our disease constellation predictions: in the case of constellation 2, its values are relatively small, and predominately contain integumentary and infectious diseases (0.016 for GLM and 0.055 for GBM). In the case of constellation 6, the values are large, and include neuropsychiatric and cardiovascular diseases (0.160 for GLM and 0.177 for GBM). As a final test, we compared these results against those for the selected five single diseases, cognizant that asthma, allergic rhinitis, depression, general hypertension and osteoarthritis can be assigned to constellations 3, 3, 10, 6 and 6, respectively (Methods). We observed that, as shown in Supplementary Table 1, our models perform better in predicting disease constellations than in predicting single diseases (except for allergic rhinitis).

## Discussion

Here, we described our efforts towards understanding diseases jointly, accounting for their similarities and differences.

Our study contains several limitations and we could have made a few different decisions. For example, we could have attempted to identify a space transformation that would maximize orthogonal dimension genetic loading, although how best to define and implement such a transformation is yet unclear. Furthermore, we could have modeled disease dimensions with a hyperbolic rather than a Euclidean metric to account for disease hierarchy[37]. Practical considerations drove our choice of 20 dimensions; with 547 discrete diseases to be represented in the space, we reasoned that the target space dimensionality should be lower than 50, based on the cross-validation identification of optimal dimensions for natural language contexts. In a perfect world, the choice of dimensionality for the disease embedding space would be treated as a model selection problem and decided upon using unambiguous optimization criteria. In practice, the embedding algorithms[12,13] are not fully suited for such rigorous model selection; rather, they rely on human-curated datasets for benchmarking—in our case, physicians' diagnostic judgment. In addition, the hypothesis of chronological co-occurrence of diseases being biologically linked may not always hold in daily clinical practice. When complete medical histories of patients ('cradle-to-grave') are available, even long-term disease associations can be captured, at least in theory. In practice, we must work with shorter, partial medical histories, and some of the longer term associations may appear lost.

Genetic pleiotropy is believed to be ubiquitous in a human genome. Geneticists study one or a few diseases as independent entities at a time, and then look for overlap in significant associations of genetic variants between pairs of diseases. Our disease embedding approach captures complex similarities among diseases and considers all diseases, putatively offering a new look at pleiotropy. Every dimension of our 20-dimensional disease space may encode the shared genetic, environmental and genetic–environmental etiology of multiple diseases. By performing GWASs for each dimension, we can explore the pleiotropic effects underlying a large group of related diseases. Our analyses include the following two analysis levels, embedding dimension and disease proximity. At the embedding dimension GWAS analysis level, we found multiple dimensions were associated with the same genetic marker variation. For example, marker *rs34290285* (near the locus *D2HGDH*) associated with dimensions 1 and 7. The locus harboring gene *HLA-B* (genetic markers *rs12212594*, *rs28380903*, *rs9265745*, *rs2428494*, *rs2523621* and *rs2523616*) associated with dimensions 1, 3, 4, 7, 12 and 16 (Fig. 4c). We then identified diseases that loaded most heavily, either positively or negatively, in these dimensions and thus we considered those the most relevant diseases. For example, autosomal abnormality, chromosomal anomaly and other developmental conditions are loaded on dimension 1. Similarly, CNS infections, septicemia and many other infectious diseases are loaded on dimension 3, while diabetes mellitus and many other endocrine diseases are loaded on

**Table 2 | Performance of polygenic prediction models**

| Disease constellation | Performance index | Embedding model: GLM | Embedding model: GBM |
|---|---|---|---|
| 1 | Nagelkerke $R^2$ (P-value) | 0.047±0.001 (<10⁻¹⁶) | 0.072±0.001 (<10⁻¹⁶) |
| | Prediction accuracy[a] | 56.6%±0.1% | 56.8%±0.1% |
| | PPV[b] | 56.3%±0.1% | 56.4%±0.1% |
| | NPV[c] | 56.9%±0.1% | 57.3%±0.2% |
| 2 | Nagelkerke $R^2$ (P-value) | 0.016±0.001 (1.6×10⁻³) | 0.054±0.001 (<10⁻¹⁶) |
| | Prediction accuracy | 52.3%±0.1% | 53.2%±0.2% |
| | PPV | 52.3%±0.1% | 53.7%±0.2% |
| | NPV | 52.4%±0.1% | 52.8%±0.1% |
| 3 | Nagelkerke $R^2$ (P-value) | 0.037±0.001 (<10⁻¹⁶) | 0.064±0.001 (<10⁻¹⁶) |
| | Prediction accuracy | 55.4%±0.1% | 55.6%±0.1% |
| | PPV | 55.3%±0.1% | 56.1%±0.1% |
| | NPV | 55.5%±0.1% | 55.3%±0.1% |
| 4 | Nagelkerke $R^2$ (P-value) | 0.036±0.001 (<10⁻¹⁶) | 0.062±0.001 (<10⁻¹⁶) |
| | Prediction accuracy | 55.3%±0.1% | 55.4%±0.1% |
| | PPV | 55.2%±0.1% | 55.1%±0.2% |
| | NPV | 55.5%±0.1% | 55.8%±0.2% |
| 5 | Nagelkerke $R^2$ (P-value) | 0.053±0.001 (<10⁻¹⁶) | 0.077±0.001 (<10⁻¹⁶) |
| | Prediction accuracy | 57.1%±0.1% | 57.3%±0.1% |
| | PPV | 56.9%±0.1% | 57.5%±0.1% |
| | NPV | 57.3%±0.1% | 57.1%±0.1% |
| 6 | Nagelkerke $R^2$ (P-value) | 0.160±0.001 (<10⁻¹⁶) | 0.176±0.001 (<10⁻¹⁶) |
| | Prediction accuracy | 64.4%±0.1% | 64.5%±0.1% |
| | PPV | 63.6%±0.1% | 63.9%±0.2% |
| | NPV | 65.4%±0.2% | 65.3%±0.2% |
| 7 | Nagelkerke $R^2$ (P-value) | 0.047±0.001 (<10⁻¹⁶) | 0.072±0.001 (<10⁻¹⁶) |
| | Prediction accuracy | 56.4%±0.1% | 56.5%±0.2% |
| | PPV | 56.1%±0.1% | 56.0%±0.2% |
| | NPV | 56.8%±0.2% | 57.2%±0.2% |
| 8 | Nagelkerke $R^2$ (P-value) | 0.033±0.001 (<10⁻¹⁶) | 0.060±0.001 (<10⁻¹⁶) |
| | Prediction accuracy | 55.0%±0.1% | 55.2%±0.1% |
| | PPV | 54.8%±0.1% | 55.2%±0.1% |
| | NPV | 55.2%±0.1% | 55.2%±0.1% |
| 9 | Nagelkerke $R^2$ (P-value) | 0.031±0.001 (4.5×10⁻¹⁶) | 0.059±0.001 (<10⁻¹⁶) |
| | Prediction accuracy | 54.7%±0.2% | 54.9%±0.2% |
| | PPV | 54.5%±0.2% | 54.7%±0.2% |
| | NPV | 55.0%±0.2% | 55.1%±0.2% |
| 10 | Nagelkerke $R^2$ (P-value) | 0.074±0.001 (<10⁻¹⁶) | 0.098±0.001 (<10⁻¹⁶) |
| | Prediction accuracy | 59.4%±0.1% | 59.7%±0.1% |
| | PPV | 59.2%±0.1% | 59.0%±0.1% |
| | NPV | 59.5%±0.1% | 60.6%±0.1% |

We constructed two embedding models based on the 116 SNP associations identified in this study (as described in Methods section 10): a GLM and a GBM. Here we report the 95 percent confidence intervals of their performance indices. [a]We defined prediction accuracy as the number of correctly classified samples to the total number of trials, which we based on the testing datasets. [b]PPV: the proportion of positive results (predicted as cases who had the disease of interest by polygenic prediction models) that are true positive, based on the testing datasets. [c]NPV: the proportion of negative results (predicted as controls who did not have the disease of interest by polygenic prediction models) that are true negative, based on the testing datasets.

dimension 16. At the disease–disease embedding proximity level, we examined a disease's closest space neighbors forming a disease cluster (constellation) and their corresponding constellation-specific GWAS loading. For example, we found that asthma's closest neighbors in the space included allergic rhinitis, emphysema chronic obstructive pulmonary disease, chronic upper respiratory infection, chronic sinusitis,

food allergy, alveolar disease and pneumonia, which all belonged to constellation 3. In the disease space, the 20-dimensional coordinates of asthma are {1.31, −0.452, −0.938, 0.197, −1.30, −0.426, 2.60, −1.26, 0.994, −1.15, −0.373, 0.969, 0.781, −0.295, −1.42, −0.897, 0.919, −0.350, −0.00992, 0.00486}. For asthma, dimensions 7, 15, 1, 5 and 8 loaded the largest absolute values, and thus we expected those values to contain

a significant amount of information relevant to asthma. Indeed, our GWASs of these five dimensions' associated genetic loci (such as *HLA-B*) to asthma and its constellation neighbors, such as allergic rhinitis, confirmed by matching them against the GWAS Catalog. These allergic diseases constellation-linked loci often harbored the genes that encode proteins regulating immunity-related pathways, such as interleukin 1 receptor activity control and interleukin 1 family signaling (Supplementary Data 6). These associations align well with our current knowledge about the etiology of asthma and its constellation neighbors. This two-pronged analysis is our blueprint for identifying the pleiotropic genetic variations shared by constellations of etiologically similar diseases.

A disease–disease closeness in the embedding space means that the diseases involved occur in similar contexts of other pathologies in the diagnostic histories of many patients. Therefore, this proximity may indicate shared disease etiology. The space encodes a measure of similarity among diseases and thus allows us to re-examine the established nosology. For instance, in the International Disease Classification (version 9; ICD-9), migraine is grouped closely with eye inflammation in the cluster of 'diseases of the CNS and sensory organs' (see ICD-9 codes 320–389). Alternatively, data-driven studies suggested that migraine should be closer to immune system diseases, such as IBS[14,38]. Using the disease space, we estimated the cosine similarities between migraine and eye inflammation, and between migraine and IBS. Indeed, we found that migraine–IBS and migraine–eye inflammation similarity values were 0.48 and 0.08, respectively, suggesting that migraine is, indeed, relatively closer to IBS. Our analysis of 20-dimensional vectors representing these three diseases showed that dimension 8 contributed most to the (dis)similarity among them. Along this dimension, migraine and IBS had the largest positive coordinate values compared with all the two diseases' other coordinates. The position of eye inflammation along this dimension, however, had the largest negative coordinate value.

Traditionally, disease taxonomies have gravitated towards grouping diseases by topographical, anatomical, institutional and cultural similarities. As such, taxonomies have inevitably incorporated often arbitrary, culturally relative and/or subjective disease groupings[14]. Relying on the distances between diseases measured in this disease embedding space, we inferred disease classification in an objective, data-driven way that inscribes the disease histories of more than a hundred million patients (Extended Data Fig. 2).

Our disease embedding method enables an integrative, system-level representation of human health state dynamics, as opposed to the traditional, reductionist, one-disease-at-a-time approach (see Supplementary Fig. 1 for the overall workflow of this study). Though distinct in methods and properties, our disease embedding approach bears an analogy to the holistic description of human health states and transitions used in holistic medical traditions, such as traditional Chinese medicine[39,40]. We hope that our disease embeddings can be useful in a variety of biomedical applications. For example, it represents massive amounts of information about shared disease etiologies and consequences, enabling the creation of clinical warning signs and the imputation of disease heritabilities and between-disease genetic correlations from an incomplete set of past estimates[16]. Because our disease embedding space provides a contextualization of patients' complex histories, when it is combined with an individual's genetic information, the space delivers more precise predictions of which treatment options may be more or less beneficial.

Our approach to computing polygenic risk scores across disease space dimensions can lead to the design of more powerful models for forecasting individual-specific health problems. In this study, we followed both the traditional, one-disease and the disease region prediction routes. The latter route—forecasting a collection of related conditions all at once, such as a collection of bronchial and lung inflammation diseases instead of asthma by itself—might prove even more productive in the future.

## Methods

### Description of utilized databases

We used four large, anonymized cohorts: the Merative MarketScan, UK Biobank, BioBank Japan and BioVU databases (see Table 1 for their sample characteristics).

The MarketScan claims database contains the United States (US) country-scale collection of diagnosis records for over 151 million unique individuals enrolled during the years between 2003 and 2013 (ref. 16).

This dataset includes a collection of insurance-claim-based records documenting inpatient and outpatient medical events, medical procedures, medications and healthcare expenditures. These data were collected from numerous private insurance companies, managed care organizations, health plan providers and state Medicaid agencies. The insured patient population is therefore biased towards more affluent, privately insured segments of US society[15]. The MarketScan database provides our analysis with three distinct strengths: (1) comprehensive diagnoses, procedure and prescription coding, at the individual patient level with a day-level temporal resolution of events; (2) the large sample size covers over half of the US population, and; (3) it contains full integration of inpatient and outpatient events, emergency care services and outpatient pharmaceutical data. There have been more than 900 peer-reviewed publications since the launch of these databases in 1995, and this number has increased even more rapidly in recent years[41].

UK Biobank (UKB) is a national health service registry database based in the United Kingdom (UK), including around 500,000 participants aged 40–69 years, and recruited between 2006 and 2010. For this study, we selected those individuals in the white British ancestry subset who had diagnosis records and genotype data available. Diagnosis records were retrieved from self-reports and medical assessments during regular visits, and this information was used to compute a mean disease embedding score for each individual. Related to these mean score values, we aimed to discover their associated genetic variants. In this regard, a total of around 96 million genetic variants, including genotyped and imputed ones, were testable.

BioBank Japan (BBJ) is a patient-based registry in Japan that describes over 267,000 participants of East Asian descent. The BBJ project was launched in 2003 to implement personalized medicine and is being conducted in three five-year periods. The diagnoses focused on a total of 51 targeted common diseases that covered 15 broad categories. These diseases were selected owing to their clinical importance in helping to largely explain morbidity or mortality in Japan. In addition, DNA samples were collected and genotyped or sequenced for genomic analyses through the cooperation of 12 medical institutes, consisting of 66 hospitals[25,27]. Previous studies have suggested that BBJ can represent Japan's general patient population after comparing it against other Japanese databases and revealing largely consistent trends in common clinical variables[26]. Here, we leveraged this dataset to replicate the significant genetic associations found in the UKB.

The BioVU database is Vanderbilt University Medical Center (VUMC)'s de-identified DNA biobank, which houses DNA samples from more than 250,000 individuals. DNA samples were collected from routine clinical testing and have been undergoing genome-wide genotyping with arrays including the Multi-Ethnic Global (MEGA) array batch by batch. The genotypes were then imputed and phased according to the Human Haplotype Reference Consortium reference panel (version r.1.1). The clinical information, including electronic medical records (EMRs), has been continuously updated as well. Again, for the purpose of replication analysis, we selected a cohort from this database which contains 16,545 individuals of European descent.

### Disease embeddings

We subjected MarketScan claims data, whose collection of diagnosis records is the largest among all the databases available to us, to this analysis. First of all, we mapped International Classification of Diseases versions 9 and 10 (ICD-9 and ICD-10) codes into 547 major disease diagnosis

groups based on clinical manifestations. These disease groups constituted the basic 'word vocabulary' upon which diagnosis records were built. To compute disease embeddings, by analogy with word embeddings, we used the word2vec algorithm[12,13]. To do this, we made the following customizations to the algorithm: (1) we replaced natural language words with disease groups; (2) we replaced sentences with chronological sequences of patient-specific disease groups, and; (3) we replaced the text corpus with a large collection of patient-specific diagnosis records.

With these customizations in place, our representation for a focal disease, $\omega$, is capable of capturing its context (that is, co-occurring diseases $\omega_-$) in a patient's diagnosis record and its relation with $\omega_-$. Such a goal can be mathematically measured using a cost function as

$$\text{cost} = -\mathscr{L} = -\sum_{\omega \in C} \log P(\omega, |, \omega_-), \qquad (1)$$

where $\mathscr{L}$ is the logarithm of likelihood and $C$ represents our 'corpus', that is, the records of 547 major diseases for over 151 million unique patients. This corpus $C$ was used to train a neural network model in conjunction with the Gensim package[42], and the context size of disease codes was set to be eight. We set the dimensionality of the disease (word) vector to be 20. For ease of disease presentation and obtaining an orthogonal coordinate system, we further applied principal component analysis to rotate the disease vectors so that dimension 1 corresponded to the first principal component, dimension 2 to the second PC, and so on, finally, dimension 20 to the twentieth PC.

The resulting disease embedding space has 20 dimensions, or in other words, each disease is represented by a 20-dimensional vector $\{E_1^\omega, \cdots, E_i^\omega, \cdots, E_{20}^\omega\}$ (see Fig. 1 for various visualizations of the disease embeddings). We chose 20 dimensions, because: (1) in principle, the dimension of the embedding space should, on one hand, be much smaller than the 'vocabulary' size (547 disease groups in our case), but on the other hand, be sufficiently large to maintain a reasonable prediction accuracy, and; (2) physicians in our team agreed that the space with 20 dimensions did, indeed, generate a reasonable nosology. In addition, we also tried different numbers of disease space dimensions, such as 50 and 300, to confirm that diseases can be distanced relative to each other in a consistent way among different space dimension choices. We computed the Euclidean distances (or cosine similarities) between any two diseases out of over 547 diseases, of which embedding values were expressed as 20-dimensional, 50-dimensional and 300-dimensional vectors, respectively. Then, we compared the Euclidean distances (or cosine similarities) measured in the 20-dimensional space with those measured in the 50-dimensional space, confirming their significantly high concordance: Pearson's correlation coefficient $\gamma = 0.74$ (or 0.57), Student's $t$-test two-sided $P < 2.2 \times 10^{-16}$; the $P$-values of regression coefficient and intercept in linear approximation were both smaller than $2.2 \times 10^{-16}$, out of Student's $t$-test to indicate a significance level of the estimates being different from 0 (see Supplementary Fig. 14a,c for comparison details). Similarly, we also observed high concordance between the Euclidean distances (or cosine similarities) measured in the 50-dimensional space and those measured in the 300-dimensional space: Pearson's correlation coefficient $\gamma = 0.62$ (or 0.59), Student's $t$-test two-sided $P < 2.2 \times 10^{-16}$; the $P$-values of regression coefficient and intercept in linear approximation were both smaller than $2.2 \times 10^{-16}$ (see Supplementary Fig. 14b,d).

### Annotation for embedding dimensions by the most separable disease categories

Each dimension of the disease embedding space consists of embedding values for 547 unique diseases that can be grouped into 21 functional categories, and we wanted to know which pair of categories were most separable. It would indicate that this dimension was the most informative in terms of differentiating the category pair which in return could be used as the dimension's annotation. To this end, we used the Wilcoxon rank sum test to examine whether and how much

the distribution of embedding values in one category is different from another[20–22]. The procedures are as follows:

(1) In each dimension, pick two disease categories A and B for investigation;
(2) Test whether the distribution of disease embedding values in B is significantly different from that in A. The bulk difference between the two distributions can be quantified by the median of the difference between a randomly selected embedding value from category B and a value from A[43];
(3) Repeat steps 1 and 2 for all the possible two-category combinations out of 21 in total and for all the 20 embedding dimensions (that is, a total of $\binom{21}{2} \times 20 = 4200$ comparisons). We controlled the FDR and adjusted all tests $P$-values using the Benjamini–Hochberg procedure[30,44] (see Supplementary Data 1 for the complete test results).

Finally, to annotate each embedding dimension, we used the pair of disease categories that showed the smallest $P$-value in this test.

### Learning constellations of disease embeddings using a $k$-SVD approach

To obtain a thematic understanding of the 20-dimensional embedding values of 547 unique diseases, we implemented a $k$-SVD approach[23,24] and generated ten disease constellations. The $k$-SVD approach, as a generalized $k$-means clustering method, originated as a dictionary learning algorithm for obtaining a dictionary for sparse representations of signals. It performs SVD to update the constellations of the dictionary one by one for a better fit to the signals. In the field of document modeling, those constellations can be considered as coherent 'word clusters' detected in document texts, while in our use case, they are reminiscent of disease clusters that tend to co-occur in diagnosis records. Given any collection of diseases, we can assign them to their most similar disease constellations, and in Fig. 3a and Supplementary Data 3, we show these assigned diseases in each constellation.

In addition, it is worth noting that we were aware of the alternatives of clustering methods, but eventually decided to use $k$-SVD method, because it had a proven success of applying to word embedding methods particularly[23]. Nevertheless, as a comparison, we tried to implement hierarchical clustering methods. We used a cosine similarity to measure a between-disease distance, and then computed average similarities between pairs of clusters (constellations, see Supplementary Data 2). We then compared ten constellations produced by hierarchical clustering and by the original $k$-SVD method. We used a hypergeometric enrichment test to probe similarities of results produced by these two methods. We found that: (1) there were one-to-one correspondences between the two sets of constellations; (2) the resulting $P$-values measuring the probability to obtain this correspondence by chance were $3.2 \times 10^{-9}$, $2.5 \times 10^{-34}$, $1.1 \times 10^{-3}$, $1.9 \times 10^{-25}$, $2.8 \times 10^{-7}$, $6.4 \times 10^{-21}$, $2.1 \times 10^{-27}$, $9.5 \times 10^{-16}$, $1.7 \times 10^{-15}$ and $6.5 \times 10^{-4}$ for the constellations from the first to tenth constellation, respectively, suggesting the clustering results are robust against different clustering methods.

### Disease embedding dimensions for individuals

With the embedding vector $\{E_1^\omega, \cdots, E_i^\omega, \cdots, E_{20}^\omega\}$ developed for each disease $\omega$ (Methods), we could then calculate a 20-dimensional vector of weighted mean disease embedding scores $\{\bar{E}_1, \cdots, \bar{E}_i, \cdots, \bar{E}_{20}\}$ for any individual whose diagnosis record, $W$, is known, using the following formula:

$$\bar{E}_i = \sum_{\omega \in W} (n_\omega E_i^\omega) / \sum_{\omega \in W} n_\omega, \qquad (2)$$

where $n_\omega$ is the count of disease $\omega$ within the record $W$.

In this way, we extended the calculations for all the participants in the three databases (UKB, BBJ and BioVU). In essence, these weighted mean scores suggest the coordinates of individuals in the

20-dimensional embedding space, and by treating them as quantitative traits, we could perform genome-wide association analyses (Methods).

## Genome-wide association analyses of embedding dimensions

To discover genetic associations related to the individual-specific embedding coordinates, we selected unrelated individuals of white British background in the UKB, whose high-quality genotype data and diagnosis records were both available; the resultant population was 306,629. As for testable SNP quality controls, we imposed the following thresholds: minor allele frequency (MAF) > 0.01 and the Hardy-Weinberg equilibrium $P > 10^{-6}$.

For each of the 20 dimensions of individual-specific disease embedding coordinates, we tested its association with additive SNP effects (that is, 0, 1 and 2 allele dosage coding) across the genome using a linear regression model[45]. We used sex, age and the first ten genetic PCs as covariates in modeling. As a result, we found 116 SNP–embedding associations to be significant, genome-wide ($P < 5 \times 10^{-8}$). There are 108 unique lead loci involved in these associations, and 40 loci therein were never reported in any historical GWASs. The novelty of these lead loci was determined via the following two steps: (1) downloaded the most updated GWAS Catalog containing a summary of historical GWAS results, and the downloaded file was updated on 17 May 2022; (2) for each of the 108 lead loci that involved in our identified associations, we tried to search within its neighborhood (±5,000 base pairs) for any loci in the catalog that have $r^2$ value (a measure of linkage disequilibrium with respective to the lead loci) greater than 0.1. If such loci cannot be found, then we would claim the novelty of the lead loci of interest. The next section describes the replication of the significant associations in the independent cohorts (see Fig. 4, and Supplementary Data 4 for summary statistics).

## Replication analyses using the BBJ and BioVU datasets

To replicate the genome-wide significant associations discovered in the UKB, we leveraged another two independent cohorts, that is, a Japanese cohort from the BBJ and an American cohort from BioVU.

First, from the BBJ, we selected a total of 166,612 individuals who had both diagnosis records and high-quality genotyping data. Just like in the discovery analysis, we adopted a multivariate regression model adjusted for sex, age and the first ten genetic PCs. We then tested the individual-specific embedding coordinates in one dimension at a time. Because an association replicates only if the sign of its regression coefficients (the natural logarithm of the odds ratios) matches between the discovery and replication results, we used one-sided $P$-values to test the replication[46]. Among the 116 discovered associations, 16 associations involved 15 unique loci that were not genotyped or imputed in the BBJ, therefore leaving 100 associations subjected to the replication attempts here. Compared to these 100 discovered associations, we found 87 in this replication analysis in the matched direction (sign) of the estimated effect. Considering that an association replicates only if the association direction matches between the discovery and replication results, we computed the one-sided $P$-value to test a replication, with an expected association direction determined by the discovery analysis[46,47]. We then adjusted $P$-values via the Benjamini–Hochberg procedure in consideration of multiple testing[30,44], and defined a successful replication at a 5% FDR. As a result, 41 associations remain significant, and thus, they were successfully replicated. We acknowledge that ethnicity difference between BBJ participants (East Asian ancestry) and the UKB participants for our GWAS discovery analyses (British white ancestry) could possibly confound the genetic association tests, since allele frequencies and linkage disequilibrium patterns of the entire genome are not all similar across different ethnic populations[48]. A slightly relieving fact is, specifically relevant to our study, historical research has shown that the overall correlations between the genetic association results concerning hundreds of different phenotypes based on European ethnic group and those based on East Asian group are very significant[49].

For an additional replication attempt, we brought in another independent dataset from BioVU by selecting 16,545 individuals of European descent, a much smaller cohort than UKB and BBJ. Again, we performed the same multivariate regression analysis for the embedding coordinates in each dimension, which we also adjusted for sex, age, the first three genetic PCs, and genotyping array types and batches. As a result, 87 of the 116 associations are in the direction consistent with our discoveries, and 15 associations are still significant, at a 5% FDR (Supplementary Fig. 3 and Supplementary Data 4).

## Pathway enrichment analysis based on GWAS summary statistics of individuals' embedding dimensions

Based on the GWAS results generated, we further investigated whether the GWAS summary statistics suggested any biological pathways or processes that were significantly enriched in an embedding-dimension-specific manner. Concerning with each dimension, we selected the lead SNPs that surpassed the suggestive threshold ($P < 10^{-5}$), and then tried to map these SNPs to genes using positional, eQTL and chromatin interaction information. With the aim of finding possible over-representation of biological pathways and biological processes, we tested these mapped genes against the 'background' gene sets that were obtained from MSigDB (including positional gene sets, curated gene sets, hallmark gene sets, gene ontology gene sets, oncogenic signatures, immunologic signatures, motif gene sets and computational gene sets)[50], and WikiPathways (19,283 protein-coding genes)[51]. We used the hypergeometric test, generating the $P$-values for each category (that is, canonical pathways and gene ontology biological processes, separately) that were further adjusted through Benjamini–Hochberg correction[52]. Finally, as reported in Supplementary Data 6, we summarized the significant findings (Benjamini–Hochberg adjusted $P < 0.05$).

## Health-related phenotypic association analyses of embedding dimensions

To examine phenotypic associations (in addition to genetic associations) across the 20 embedding dimensions, we used a collection of 140 phenotypic data entries available in the UKB resource[28] that measured ten general categories, including blood count, blood biochemistry, urine biochemistry, spirometry, early-life factors, anthropometry, addictions, diet, physical activity and sleep, and local environment (for example, the coverage of greenspace/natural environment and air quality measured by nitrogen dioxide, nitrogen oxide and particulate matter). Particularly, spirometry includes pulmonary function measures on forced expiratory volume in one second ($FEV_1$), forced vital capacity (FVC), ratio of $FEV_1$ to FVC and peak expiratory flow (PEF). We first computed their respective predicted values based on the prediction equations developed for white male and female adult participants in the third US National Health and Nutrition Examination Survey[53], and then derived the percentage predicted values through normalizing the measured against the predicted values. Finally, it is worth noting that we applied a min–max normalization to all these phenotypic measures, making their values all vary from 0 to 1, and thus the association coefficients attached to these measures could be directly compared with each other.

This analysis was conducted based on the same set of samples as used in GWAS discovery (Methods), that is, the 306,629 unrelated individuals of white British background who had both diagnosis records and high-quality genotype data available. Focusing on one phenotypic measure at a time, we tested its association with each of the 20 dimensions of individual-specific disease embedding coordinates using a multivariate regression model. Covariates in modeling included sex, age and the first 10 genetic PCs; height was additionally included, if the to-be-tested phenotypic measure concerned spirometry. This association analysis was repeated for all of the 140 phenotypic measures. The resulting association coefficient of the phenotype can characterize how the phenotype associates with a given

embedding dimension: a positive (or negative) coefficient indicates a positive (or negative) association; the greater the absolute coefficient value is, the stronger the association is. We used the Student's $t$-test to determine whether the coefficient estimates significantly differed from zero, and controlled the FDR using the Benjamini–Hochberg procedure[30,44]. An association was deemed to be significant at an FDR of 0.05, and the results are summarized in Supplementary Figs. 4–13 and Supplementary Data 7.

### Polygenic prediction model construction

Starting with the genome-wide significant findings, we made further efforts in developing polygenic prediction models in order to estimate how much the ensemble of our identified genetic variants could predict disease susceptibility.

Firstly, the prediction models were built on polygenic risk scores (PRSs), which measure the genetic liability to human complex traits by aggregating the effects of genome-wide genetic markers (usually SNPs)[54]. Thus, the PRS for the $i$-th individual, $\hat{S}_i$, can be expressed as:

$$\hat{S}_i = \sum_{j=1}^{m} X_{ij}\hat{\beta}_j, \qquad (3)$$

where $X_{ij}$ is the risk allele count for the $j$-th SNP ($j = 1, \cdots, m$) of the $i$-th individual ($i = 1, \cdots, n$), and $\hat{\beta}_j$ represents the assigned weight to each SNP. Typically, an estimated regression coefficient (the odds ratio's natural logarithm) from GWAS is used as the weight, $\hat{\beta}_j$. Related to the embedding scores of dimensions from 1 to 20, there were various numbers of independent SNPs identified (see Supplementary Table 3 for the SNP numbers). Because we based these SNP estimations—as well as their respective $\hat{\beta}_j$—on the embedding scores derived from the 547 diseases in total, we propose that they shall be universal across different diseases. For each dimension, we could compute an individual's $\hat{S}_i$ using equation (3), so that each individual has a 20-dimensional vector of $\hat{S}_i$ values.

Next, we tested these values' utility by repeatedly applying them in polygenic prediction modeling, to predict whether an individual is susceptible to certain single diseases or disease constellations.

### Single disease prediction

A given disease has its own specific set of $m$ SNPs and their respective $\hat{\beta}_j$. For example, in the case of asthma, one of the most abundant diseases recorded in the UKB, we searched the published studies in the GWAS Catalog database[31] for the summary statistics specific to asthma, locating 203 independent SNPs that were involved in 293 asthma-specific associations. So, $m = 203$, and if we encountered multiple $\hat{\beta}_j$ values for the same SNP, we used an averaged value. In the same way, we obtained the disease-specific $\hat{\beta}_j$ values and computed the individual's $\hat{S}_i$ using equation (3) for the other four most abundant diseases in the UKB: allergic rhinitis, depression, general hypertension and osteoarthritis. The respective values of $m$ for these four diseases are 29, 348, 73 and 67. This traditional way of computing $\hat{S}_i$ yielded disease-specific estimates.

We retrieved estimates of Nagelkerke $R^2$ from previously published studies, which modeled the relationship between disease-specific $\hat{S}_i$ values and disease risk using the binomial family's GLM and served as baseline data here. We then implemented three polygenic prediction models. First, we used a conventional model built on the GWAS Catalog[31]; before adopting the same GLM algorithm, we re-computed disease-specific $\hat{S}_i$ values for UKB participants using the set of SNPs and their $\hat{\beta}_j$ that were available in the GWAS Catalog database and specific to the disease of interest. Second, we used a GLM model with universal $\hat{S}_i$; settings the same as in the first model, except that now we used the universal $\hat{S}_i$ that we derived out of the association analyses against 20-dimensional embedding coordinates. Third, we used a non-linear model with universal $\hat{S}_i$; we replaced the GLM in the second model with a non-linear model, that is, a GBM[32], while retaining the other settings.

In all three models, we considered the following confounding factors: age, sex and ethnicity (40 genetic PCs). To train the models for each disease, we randomly drew 5,000 cases in which patients were diagnosed with the disease of interest and 5,000 control patients who were not. To test the model performance, another 2,500 cases and 2,500 controls were drawn from the remaining samples. To enable the estimation of the results' confidence intervals, we repeated this process of training and testing 100 times. In Supplementary Table 1, we report various performance measures with the respective 95% confidence intervals, including Nagelkerke $R^2$ and prediction accuracy. We used Nagelkerke $R^2$ to assess the goodness of model fitting to data, and computed $P$-values using the likelihood ratio test. Based on the testing datasets, we computed the prediction accuracy, and defined it as the number of correctly classified samples as compared with the total number of trials; we also computed PPV, defined as the proportion of positive results (predicted as cases who had the disease of interest by polygenic prediction models) that are true positive, and NPV, defined as the proportion of negative results (predicted as controls who did not have the disease of interest by polygenic prediction models) that are true negative.

Additionally, we reported the performance of our models for all the other diseases that had the numbers of cases more than 15,000 in the UK Biobank data (that is, at least twice of the sum of 5,000 cases for model training and 2,500 cases for model testing), and there were 25 diseases in total (Supplementary Table 2). Similar to what we concluded from the five exemplar diseases, our embedding-vector-based models can provide Nagelkerke $R^2$ values that were always greater than conventional models built on disease-specific GWAS Catalog loci, indicating that our models proposed unanimous gain in the explanatory power of genetic variation. The accuracies of discriminating cases from controls in the testing datasets offered by our models were greater than 60% for 12 of the 30 diseases we reported in total.

### Disease constellation prediction

We also predicted whether an individual would likely carry any diseases that belong to a certain disease constellation, and in other words, it is the disease constellation label that we tried to predict for an individual. To achieve this, first of all, we labeled individuals with their appropriate disease constellations by following a two-step procedure:

(1) Given the 20-dimensional embedding vectors of the 547 diseases and of the ten disease constellations, we computed each disease's cosine similarity with respect to each of the ten disease constellations. We then claimed that the disease would belong to the disease constellation with which it has the largest cosine similarity value (Supplementary Data 3). Supplementary Table 4 summarizes the 547 diseases' allocations in the ten constellations.

(2) Given a patient's diagnosis record, we would be able to label the patient with all the possible disease constellations to which the diseases in one's record belonged (as determined in the first step above). Supplementary Table 5 summarizes the constellation assignments for the 337,205 patients of white British background in the UKB (please note that a patient can have multiple disease constellation labels).

Similar to what we did for single disease prediction, we examined the predictive power of the newly constructed, universal $\hat{S}_i$ (the same ones as used in the single disease predictions) towards the disease constellation assignments for individuals using the GLM and the GBM. The confounding factors in modeling and the sample numbers for model training and testing were set the same as described above in Methods 'Single disease prediction'. We report the 95% confidence intervals of Nagelkerke $R^2$ and prediction accuracy in Table 2.

## Ethics statement

The research was approved by the University of Chicago Institutional Review Board, and passed ethical approval by the respective organizations that maintain the individual databases.

## Reporting summary

Further information on research design is available in the Nature Portfolio Reporting Summary linked to this article.

## Data availability

Source data for Figs. 1–4 and Extended Data Figs. 1 and 2 are available with this manuscript. The license for MarketScan databases is available to purchase by federal, nonprofit, academic, pharmaceutical and other researchers. Access to the data is contingent on completing a data-use agreement and purchasing the needed license. More information about licensing the MarketScan databases can be found at https://www.merative.com/documents/brief/Marketscan_explainer_general. The phenotypic and genetic datasets of UK Biobank used in this study are available via the UK Biobank data access process (http://www.ukbiobank.ac.uk/register-apply/) and detailed information can be found at http://www.ukbiobank.ac.uk/scientists-3/genetic-data/ and http://biobank.ctsu.ox.ac.uk/crystal/label.cgi?id=100314. Access to the phenotypic and genetic datasets of BioVU can be requested after a study proposal is received, approved by the BioVU Review Committee and a user agreement is signed. More information can be found at https://victr.vumc.org/how-to-use-biovu/. The availability about the phenotypic and genetic datasets of Biobank Japan is described at https://biobankjp.org/english/index.html, and more information can be found at https://humandbs.biosciencedbc.jp/en/hum0014-v21.

## Code availability

We provide a Code Ocean capsule including executable programming scripts, and input and output data[55]: https://doi.org/10.24433/CO.0096653.v1.

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

## Acknowledgements

We thank E. Gannon and M. Rzhetsky for comments on earlier versions of this manuscript, and many volunteers whose data are used in this study.

This work was funded by the DARPA Big Mechanism program under ARO contract W911NF1410333, by National Institutes of Health grants R01HL122712, 1P50MH094267 and U01HL108634-01, and by a gift from L. and K. Dauten to A.R. Additional support came from King Abdullah University of Science and Technology, award numbers FCS/1/4102-02-01, FCC/1/1976-26-01, REI/1/0018-01-01 and REI/1/4473-01-01 to X.G.; and came from Agricultural Genomics Institute at Shenzhen, Chinese Academy of Agricultural Sciences, and the Central Public-interest Scientific Institution Basal Research Fund (1102431600020230001) to G.J. The funders had no role in study design, data collection and analysis, decision to publish, or preparation of the manuscript.

## Author contributions

A.R., G.J., X.G. and Y.L. designed the study. G.J. analyzed data, performed GWAS and replication analysis, and computed polygenic risk scores. A.R. and G.J. estimated embedding from clinical data. K.W. estimated heritability and genetic correlations of 20 dimensions. G.J., Y.L., R.A., A.E., H.L. and D.E.K. built polygenic prediction models. H.K.I. and M.P. advised on the analyses involved UK Biobank data. X.Z. and N.J.C. contributed in replication analysis using BioVU data. C.T., M.A., K.M., T.G., Y.K. and M.K. contributed in replication analysis using Biobank Japan data. J.E. advised on the derivation of disease constellations and made efforts to improve the manuscript. A.R., G.J. and X.G. wrote the manuscript. All authors reviewed and approved this manuscript.

## Competing interests

The authors declare no competing interests.

## Additional information

**Extended data** is available for this paper at https://doi.org/10.1038/s43588-023-00453-y.

**Correspondence and requests for materials** should be addressed to Gengjie Jia, Xin Gao or Andrey Rzhetsky.

[1]Shenzhen Branch, Guangdong Laboratory of Lingnan Modern Agriculture, Genome Analysis Laboratory of the Ministry of Agriculture and Rural Affairs, Agricultural Genomics Institute at Shenzhen, Chinese Academy of Agricultural Sciences, Shenzhen, China. [2]Computational Bioscience Research Center, King Abdullah University of Science and Technology, Thuwal, Saudi Arabia. [3]Computer Science Program, Computer, Electrical and Mathematical Sciences and Engineering Division, King Abdullah University of Science and Technology, Thuwal, Saudi Arabia. [4]Department of Computer Science and Engineering, The Chinese University of Hong Kong, Hong Kong, People's Republic of China. [5]Department of Medicine and Vanderbilt Genetics Institute, Vanderbilt University Medical Center, Nashville, TN, US. [6]Department of Medicine, Institute of Genomics and Systems Biology, Committee on Genomics, Genetics, and Systems Biology, University of Chicago, Chicago, IL, US. [7]Department of Operations, Business Analytics, and Information Systems, University of Cincinnati, Cincinnati, OH, US. [8]Department of Systems Pharmacology and Translational Therapeutics, Perelman School of Medicine, University of Pennsylvania, Philadelphia, PA, US. [9]Extreme Computing Research Center, Computer, Electrical and Mathematical Sciences and Engineering Division, King Abdullah University of Science and Technology, Thuwal, Saudi Arabia. [10]College of Computer Science and Engineering, University of Jeddah, Jeddah, Saudi Arabia. [11]HPE HPC/AI EMEA Research Laboratory, Bristol, UK. [12]RIKEN Center for Integrative Medical Sciences, Yokohama, Japan. [13]Clinical Research Center, Shizuoka General Hospital, Shizuoka, Japan. [14]Department of Applied Genetics, The School of Pharmaceutical Sciences, University of Shizuoka, Shizuoka, Japan. [15]Department of Ophthalmology, Graduate School of Medical Sciences, Kyushu University, Fukuoka, Japan. [16]Department of Computational Biology and Medical Sciences, Graduate School of Frontier Sciences, The University of Tokyo, Tokyo, Japan. [17]Biological and Environmental Science and Engineering Division, King Abdullah University of Science and Technology, Thuwal, Saudi Arabia. [18]Department of Sociology, University of Chicago, Chicago, IL, US. [19]Department of Human Genetics, University of Chicago, Chicago, IL, US. [20]These authors contributed equally: Gengjie Jia, Yu Li. [21]These authors jointly supervised this work: Gengjie Jia, Xin Gao, Andrey Rzhetsky. ✉e-mail: jiagengjie@caas.cn; xin.gao@kaust.edu.sa; arzhetsky@uchicago.edu

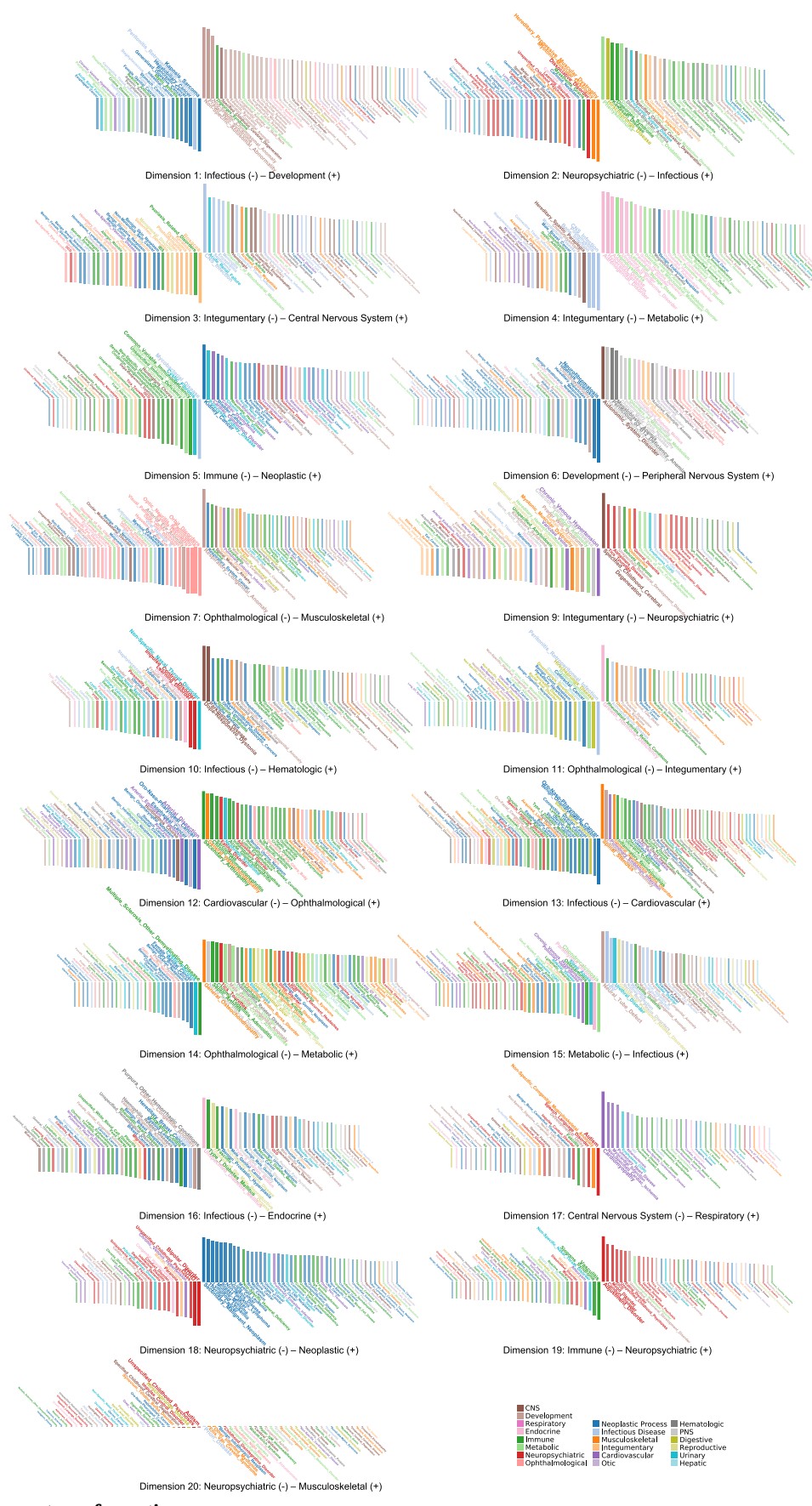

**Extended Data Fig. 1 | See next page for caption.**

**Extended Data Fig. 1 | Diseases ordered by their positions along each of the other 19 dimensions.** Related to Fig. 2c showing Diseases ordered by their positions along Dimension 8, here we show the other 19 dimensions. We rank diseases in each dimension according to the value of their corresponding coordinate. We plot positive and negative values above and below a horizontal axis, respectively. We plot the largest absolute values at the middle and other, descending values from the middle to the right for positive values, and from the middle to the left for negative ones. The bars are color-coded by the disease category.

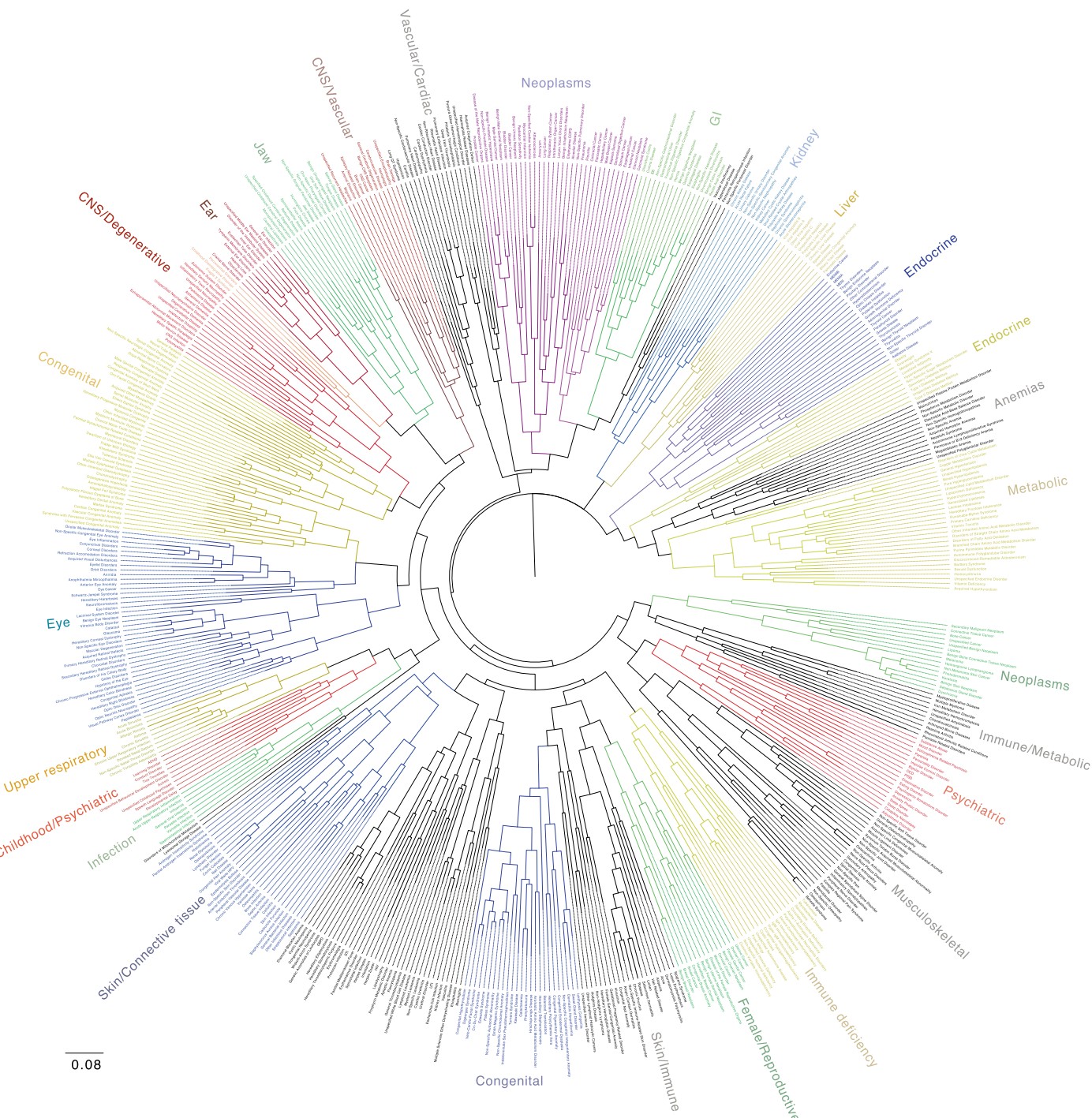

**Extended Data Fig. 2 | A disease classification based on disease embeddings.** Based on the distances between diseases measured in the disease embedding space, we can infer disease classification in an objective, data-driven way.

# Reporting Summary

## Statistics

For all statistical analyses, confirm that the following items are present in the figure legend, table legend, main text, or Methods section.

| n/a | Confirmed | |
|---|---|---|
| ☐ | ☒ | The exact sample size (*n*) for each experimental group/condition, given as a discrete number and unit of measurement |
| ☐ | ☒ | A statement on whether measurements were taken from distinct samples or whether the same sample was measured repeatedly |
| ☐ | ☒ | The statistical test(s) used AND whether they are one- or two-sided<br>*Only common tests should be described solely by name; describe more complex techniques in the Methods section.* |
| ☐ | ☒ | A description of all covariates tested |
| ☐ | ☒ | A description of any assumptions or corrections, such as tests of normality and adjustment for multiple comparisons |
| ☐ | ☒ | A full description of the statistical parameters including central tendency (e.g. means) or other basic estimates (e.g. regression coefficient) AND variation (e.g. standard deviation) or associated estimates of uncertainty (e.g. confidence intervals) |
| ☐ | ☒ | For null hypothesis testing, the test statistic (e.g. *F*, *t*, *r*) with confidence intervals, effect sizes, degrees of freedom and *P* value noted<br>*Give P values as exact values whenever suitable.* |
| ☐ | ☒ | For Bayesian analysis, information on the choice of priors and Markov chain Monte Carlo settings |
| ☒ | ☐ | For hierarchical and complex designs, identification of the appropriate level for tests and full reporting of outcomes |
| ☐ | ☒ | Estimates of effect sizes (e.g. Cohen's *d*, Pearson's *r*), indicating how they were calculated |

*Our web collection on statistics for biologists contains articles on many of the points above.*

## Software and code

Policy information about availability of computer code

| Data collection | No software was required for data collection (data preexisted and were provided by third parties). |
|---|---|
| Data analysis | We used R 4.1 and Python 3.8 to write our scripts.  We provide a Code Ocean capsule including executable programming scripts, and input and output data at  Jia, G. et al. The high-dimensional space of human diseases built from diagnosis records and mapped to genetic loci. Code Ocean https://doi.org/10.24433/CO.0096653.v1 (2023). |

For manuscripts utilizing custom algorithms or software that are central to the research but not yet described in published literature, software must be made available to editors and reviewers. We strongly encourage code deposition in a community repository (e.g. GitHub). See the Nature Portfolio guidelines for submitting code & software for further information.

## Data

Policy information about <u>availability of data</u>

All manuscripts must include a <u>data availability statement</u>. This statement should provide the following information, where applicable:

- Accession codes, unique identifiers, or web links for publicly available datasets
- A description of any restrictions on data availability
- For clinical datasets or third party data, please ensure that the statement adheres to our <u>policy</u>

Source data for Figs. 1-4 and Extended Data Figs. 1-2 are available with this manuscript. The license of MarketScan databases is available to purchase by Federal, nonprofit, academic, pharmaceutical, and other researchers. Access to the data is contingent on completing a data use agreement and purchasing the needed license. More information about licensing the MarketScan databases can be found at https://www.merative.com/documents/brief/Marketscan_explainer_general. The phenotypic and genetic datasets of UK Biobank used in this study are available via the UK Biobank data access process (see http://www.ukbiobank.ac.uk/register-apply/), and detailed information can be found at http://www.ukbiobank.ac.uk/scientists-3/genetic-data/ and http://biobank.ctsu.ox.ac.uk/crystal/label.cgi?id=100314. Access to the phenotypic and genetic datasets of BioVU can be requested after a study proposal is received, approved by the BioVU Review Committee and a user agreement is signed. More information can be found at https://victr.vumc.org/how-to-use-biovu/. The availability about the phenotypic and genetic datasets of Biobank Japan is described at https://biobankjp.org/english/index.html, and more information can be found at https://humandbs.biosciencedbc.jp/en/hum0014-v21.

## Human research participants

Policy information about <u>studies involving human research participants and Sex and Gender in Research.</u>

| Reporting on sex and gender | In all cohorts (MarketScan, BioVU, UK BioBank, BioBank Japan) we used the full gamut of populations characteristics, including sex and gender. |
|---|---|
| Population characteristics | We used sex, age, diagnostic data, clinical tests, and genotypic data when available. |
| Recruitment | Recruitment is not applicable because we used the existing de-identified data. |
| Ethics oversight | The research was approved by the University of Chicago Institutional Review Board, and passed ethical approval by the respective organizations that maintain the individual databases. |

Note that full information on the approval of the study protocol must also be provided in the manuscript.

# Field-specific reporting

Please select the one below that is the best fit for your research. If you are not sure, read the appropriate sections before making your selection.

☒ Life sciences ☐ Behavioural & social sciences ☐ Ecological, evolutionary & environmental sciences

For a reference copy of the document with all sections, see nature.com/documents/nr-reporting-summary-flat.pdf

# Life sciences study design

All studies must disclose on these points even when the disclosure is negative.

| Sample size | Sample size of each cohort was pre-determined by the data providers, therefore no special efforts were required in this regard. |
|---|---|
| Data exclusions | No data were excluded from the analyses. |
| Replication | Our replication strategies was to use UK Biobank as a discovery cohort, and BioBank Japan and BioVU as replication cohorts. |
| Randomization | Not applicable to our study design: randomization is applicable to clinical trials, but not required in observational studies. |
| Blinding | Not applicable to our study design: we did not assign treatment/placebo pairs as is done in clinical trials, but run observational association studies. |

# Reporting for specific materials, systems and methods

We require information from authors about some types of materials, experimental systems and methods used in many studies. Here, indicate whether each material, system or method listed is relevant to your study. If you are not sure if a list item applies to your research, read the appropriate section before selecting a response.

## Materials & experimental systems

| n/a | Involved in the study |
|-----|----------------------|
| ☒ | Antibodies |
| ☒ | Eukaryotic cell lines |
| ☒ | Palaeontology and archaeology |
| ☒ | Animals and other organisms |
| ☒ | Clinical data |
| ☒ | Dual use research of concern |

## Methods

| n/a | Involved in the study |
|-----|----------------------|
| ☒ | ChIP-seq |
| ☒ | Flow cytometry |
| ☒ | MRI-based neuroimaging |

