## [Peer Review File · Nature Computational Science]

Peer Review Information

Journal: Nature Computational Science

Manuscript Title: The High-dimensional Space of Human Diseases Built from Diagnosis Records and Mapped to Genetic Loci

Corresponding author name(s): Professor Andrey Rzhetsky, Genjie Jia, and Xin Gao

Reviewer Comments & Decisions:

Decision Letter, initial version:

Date: 3rd March 22 10:10:24

Last Sent: 3rd March 22 10:10:24

Triggered By: Ananya Rastogi

From: ananya.rastogi@nature.com

To: arzhetsky@uchicago.edu

BCC: ananya.rastogi@nature.com

Subject: Decision on Nature Computational Science manuscript NATCOMPUTSCI-21-0888

Message: ** Please ensure you delete the link to your author homepage in this e-mail if you wish to forward it to your co-authors. **

Dear Professor Rzhetsky,

Your manuscript "The Continuous Space of Human Diseases Mapped to Genetic Loci Predicts Disease Trajectories and Risk" has now been seen by 3 referees, whose comments are appended below. You will see that while they find your work of interest, they have raised points that need to be addressed before we can make a decision on publication.

The referees' reports seem to be quite clear. Naturally, we will need you to address all of the points raised.

While we ask you to address all of the points raised, the following points need to be substantially worked on:

- Please provide the rationale behind the assumption of the transferability of true GWAS association signals.
- It is mentioned that 547 diseases were studied but the 116 GWAS association signals were shown for only 5 diseases. As requested by Reviewer #1, please show

the results for all diseases.

- Please provide more details about the 20 dimensions that summarize 547 diseases and the reason for choosing 20 dimensions for this representation.
- As mentioned by Reviewer #2, please justify the assumptions in the study such as disease space being underpinned by genetic space and overlooking environmental effects. Also, discuss the impact of demographics, environment, lifestyle and social deprivation.
- In the Results section, please explore the meaning of the observed associations.
- As pointed out by Reviewer #3, the strategy used to define the broader defined disease (i.e. phenotypes) using ICD taxonomies should be discussed.
- Given the similarity in the fields, previous literature in the field of topological data analysis approaches needs to be discussed.

Please use the following link to submit your revised manuscript and a point-by-point response to the referees' comments (which should be in a separate document to any cover letter):

** This url links to your confidential homepage and associated information about manuscripts you may have submitted or be reviewing for us. If you wish to forward this e-mail to co-authors, please delete this link to your homepage first. **

To aid in the review process, we would appreciate it if you could also provide a copy of your manuscript files that indicates your revisions by making use of Track Changes or similar mark-up tools. Please also ensure that all correspondence is marked with your Nature Computational Science reference number in the subject line.

In addition, please make sure to upload a Word Document or LaTeX version of your text, to assist us in the editorial stage.

If you have any issues when updating your Code Ocean capsule during the revision process, please email the Code Ocean support team Cc'ing me.

To improve transparency in authorship, we request that all authors identified as 'corresponding author' on published papers create and link their Open Researcher and Contributor Identifier (ORCID) with their account on the Manuscript Tracking System (MTS), prior to acceptance. ORCID helps the scientific community achieve unambiguous attribution of all scholarly contributions. You can create and link your ORCID from the home page of the MTS by clicking on 'Modify my Springer Nature account'. For more information please visit <http://www.springernature.com/orcid>.

We hope to receive your revised paper within three weeks. If you cannot send it within this time, please let us know.

Best regards,

Ananya Rastogi, PhD
Associate Editor

Nature Computational Science

Reviewers comments:

Reviewer #1 (Remarks to the Author):

Stuy by G. Jia and colleagues:

1-Title: please consider replacing 'The Continuous Space of Human Diseases Mapped to Genetic Loci Predicts 2 Disease Trajectories and Risk' by 'The High Dimensional Space of Human Diseases Mapped to Genetic Loci Predicts 2 Disease Trajectories and Risk'.

2-Abstract: 'We validated these discovered associations using two independent cohorts: BioBank Japan and BioVU from Vanderbilt University in the US'. I think the authors make an overstatement here. 8 out of 116 GWAS associations identified in the UK Biobank replicate both in BioBank Japan and BioVU, using a liberal threshold of 0.05 (FDR). In fact only 6.5% of the associations replicate in independent samples, which may rather argue in favor of random replication of an initial false positive association (Dahl et al., Genetics 2019).

3-Results: 'In order to computationally construct this disease space, we considered each patient's health history, chronologically-ordered such that diseases experienced concurrent with one another (or soon before or after) share each other as context'. The hypothesis of chronological co-occurrence of diseases that are biologically linked is not necessarily what we observe in clinics. As an illustration, hypothyroidism can manifest twenty years after the onset of type 1 diabetes in certain patients, even though autoimmunity clearly drives the two disorders.

4-Results: the authors may consider adding more details about the 20 dimensions that summarize 547 diseases. What did capture these 20 dimensions that may help us to understand the general mechanisms at work in disease physiopathology (e.g. brain development, inflammation, immunity, nutrition, oncogenesis...)?

5-Results: as the authors did not describe the biological mechanisms underlying the 20 dimensions, it is difficult to make sense of the 116 GWAS associations identified for these 20 outcomes.

6-Results: using the Japan Biobank (East Asian ancestry) to replicate GWAS associations from the UK Biobank (European ancestry) is highly questionable. Ethnic transferability of true GWAS association signals has been shown to be only partial in literature.

7-Results: estimating the predictive utility of the 116 GWAS association signals in only 5 diseases seems quite arbitrary, as the initial analysis included 547 diseases. We cannot exclude that the authors picked-up diseases that work the best. Summarizing the prediction ability for all diseases may make more sense.

Reviewer #2 (Remarks to the Author):

The authors present an interesting study of the "human disease space" by organising co-occurring diseases in shared constellations. They identify some potentially valuable genetic associations with disease constellations and use genetic associations to define polygenic risk prediction.

I don't think the disease networks and resulting constellations are particularly novel, although the approach is valid. However, use their use as a quantitative trait for GWAS may be novel and this is the most interesting aspect of manuscript. It would be interesting to understand the relationship between the GWAS associations to constellations and the individual constituent diseases. Particularly interesting are strong GWAS associations in loci that are not associated with constituent diseases. These may be the 44 loci described? What do these loci represent, what processes, and how might they impact disease development and multimorbidity? This is not addressed in the manuscript, but seems like an important question?

Detailed comments on the manuscript:

1) Abstract

The abstract does not cover the material very clearly and some of the claims for the methods are a little overstated, e.g.

"We show that our disease-embedding-model manifests ubiquitous power in explaining phenotypic variation from genetic variation than conventional polygenic risk score models."

This is a bold statement which isn't fully justified by the results. What is ubiquitous power for example?

Sometimes the language is a little colloquial, for example it's not clear what the authors mean by terms such as distilled in this context.

The abstract reports 44 novel associations but these are not described in the manuscript!

2) Introduction

The introduction refers briefly to the disease network concepts that are fundamental to this work, before moving on to focus on genetic theory. The objective to "transcend the arbitrariness of past schemes" is a reasonable one, but many arbitrary assumptions are still made, mainly about the role of genetics. Most notably, disease space underpinned by genetic space overlooks environment effects. How are demographics, environment, lifestyle and social deprivation accounted in this model. Could they not be included as this data is available in datasets like UK-Biobank Overall the Introduction is too short and fails to introduce most of the concepts applied in this manuscript.

2) Results

The description of autoencoders would be better placed in the introduction.

The results section is rather brief and relies heavily on the figures. Little interpretation is performed in the results or discussion. More interpretation would improve the study.

3) Figure 1

In the text, fig 1 is described as a projection of 20 dimensions based on PCA, however the projection is T-SNE. No scale is indicated and more importantly T-SNE represents local structure only, and is inappropriate for representation of global structure which is the goal of this plot.

The application of Principal component analysis is very poorly described in the methods and T-SNE is not described. Overall it was very hard to follow the process to generate the 20 dimensions. This seems like a fundamental weakness in the study and should be clarified throughout

4) Figure 2

I did not find figure 2 particularly informative, if used I suggest restricting to 2c only

5) Figure 3

Figure 3 is quite a good description of the phenotypic content of each constellation. Figure 3a would benefit from a better colour scheme. It is difficult to differentiate the colours, and the disease groupings are not always intuitive.

6) Figure 4

The figure is interesting and potentially highlights useful findings. It would be particularly interesting to explore pleiotropy. For example, there are 490 individual trait associations with HLA-B in the GWAS catalog, how do these overlap with the constituents of the constellations that show association at the HLA-B locus?

There is a lot of potential to explore pleiotropy and this might also highlight loci that show no or limited association to the traits in the constellations.

The abstract and text describing fig 4 reports that 44 loci have never been reported in other association studies, yet these are not described or evaluated? What are these loci, how is their novelty determined? These are potentially very important but completely overlooked in the manuscript.

7) Discussion

The discussion includes a reasonable consideration of the limitations of the study, but very limited depth in the consideration of the results. As discussed above, the study would benefit from more consideration of the meaning of the associations. Do they represent pleiotropy? Do they represent common disease mechanisms? Do disease groupings in the constellations have common mechanisms between diseases that are otherwise dissimilar? The HLA association seems key to me here, many diseases show HLA associations in GWAS indicating an autoimmune basis. Do the HLA associated constellations include autoimmune conditions?

Overall the study raises many questions and would be strengthened by some consideration of mechanistic insights from the analysis.

Reviewer #3 (Remarks to the Author):

Authors present an interesting approach for multiple disease presentation and test how its mapping with genetic loci. The title states this approach is able to predict disease trajectories and risk, however is not clear what authors mean by "trajectories" and their definition. Furthermore, prediction model performances are not good enough to state that the model is able to predict disease risks. I would

change the title accordingly to these comments, underling other more important and innovative aspects of the proposed approach, such as the disease space representation.

In general, also given words restriction, the paper could be better structured in order to facilitate the understanding of some fundamental steps. Figures quality is in sometimes very poor and impede a proper interpretation of the results.

Specific comments.

1. Line 103, the definition of trajectories, especially their sequential dimension, should be clearly introduced. This should be consistent across the whole paper. Note that the word "trajectories" is then only used here in line 134, but no mention to them is given in methods or results. At the end of the paper, the reader still doesn't have a clear idea of what a disease trajectory is.
2. Line 113. Which are the disease space dimension? Are they only spatial or also temporal?
3. Line 115. What do you mean by "focal disease"?
4. Line 125. While is clear that each patient is represented by a sequence of disease, is not clear how (or if) the temporal dimension has been considered.
5. Line 127. Which is the strategy in defying the broader defined disease (i.e. phenotypes) using ICD taxonomies? Have you considered approaches such as clinical classification system or PheCodes? Which could better include disease clinical/physio pathological information.
6. While limitations regarding the choice of 20 dimension are discussed later in the manuscript, at this point of the paper I would provide some additional (even empirical evidence) for this choice.
In general, several of the parameters choices could have been greatly impacted the final results, and affected the disease space representation. It would be important to compare different setting, possibly with a grid search approach.
7. Line 139. Please provide a more detailed definition of the point cloud.
8. Line 142/143. I think this refer to the t-SNE projections, as indicated in figure 1 caption. It would be better to report this here too.
9. "Our disease embedding space contradicts some intuitions formed from living in two and three dimensions." This sentence is not clear.
10. Disease proximity is represented via cosine similarity. Your approach can be compared to topological data analyses approaches (see "Identification of type 2 diabetes subgroups through topological analysis of patient similarity" by Li), not only for this specific choice, but also because its ability to represent of disorder spectrum as points in a continuous space, which is foremost important in the context of precision medicine. I would refer to these previous attempts and results, even if applied in specific diseases contexts. Some topological data analyses applications (see "Using topological data analysis and pseudo time series to infer temporal phenotypes from electronic health records" by Dagliati) can also deal with disease trajectories in time.
11. Figure 2c. These are very interesting results, however it looks a bit of "cheery picking", why have you chosen to show this dimension?
12. I would respectfully argue that SVD is a recently developed approach. Also, while k-SVD clustering might sounds if applied to your raw data, as you've already computed the embeddings there are other approaches that might work as well? Again, it might be worth to compare the topological approaches, where the clustering step is often performed via hierarchical clustering, or different clustering techniques can be compared.
13. Figure 3 (especially panel b) quality is poor and it does not allow to read results.

14. Lines 198 and line 203/204. Again, I would better clarify how the consequentially similar diseases are represented, that is disease trajectory. Line 203/204 might explain my previous point, regarding what each object (spheres in figure 1) in the disease embedding space represents. It would be worth to better explain this in the previous section. An example might facilitate the reader to have a better idea of the disease space/trajectories.

15. Figure 3 is a bit confusing. It includes constellations (at this point not yet exploited for the estimates of genetic correlation) and dimensions with their genetic correlation estimations. I think it would be clearer to include 3b and 3c with figure 4.

16. Lines 309/310. Prediction accuracy is always very low, might be interesting to include PPV and NPV to better understand the reason of poor performances.

Author Rebuttal to Initial comments

We thank the editor and reviewers for their extremely thoughtful comments and suggestions. We believe that through addressing them, we have substantially strengthened the scope and clarity of our analysis. Specifically, in response to the reviews, we performed — and now report — on a battery of additional analyses. In the marked-up version, these substantial changes appear in red. We also now provide an easier-to-read version of the manuscript with the markup removed. Our point-by-point responses to reviewers are shown below, in blue.

Reviewers' comments:

Reviewer #1 (Remarks to the Author):

Study by G. Jia and colleagues:

1-Title: please consider replacing 'The Continuous Space of Human Diseases

Mapped to Genetic Loci Predicts Disease Trajectories and Risk' by 'The High Dimensional Space of Human Diseases Mapped to Genetic Loci Predicts Disease Trajectories and Risk'.

R1-1:

Following the reviewer's and third reviewer's suggestions (see R3-0-1), we have changed the title of our manuscript to 'The High-dimensional Space of Human Diseases Built from Diagnosis Records and Mapped to Genetic Loci'

2-Abstract: 'We validated these discovered associations using two independent cohorts: BioBank Japan and BioVU from Vanderbilt University in the US'. I think the authors make an overstatement here. 8 out of 116 GWAS associations identified in the UK Biobank replicate both in BioBank Japan and BioVU, using a liberal threshold of 0.05 (FDR). In fact, only 6.5% of the associations replicate in independent samples, which may rather argue in favor of random replication of an initial false positive association (Dahl et al., Genetics 2019).

R1-2:

The reviewer raises an important and valid point.

There are at least three major hurdles involved with replicating genetic associations:

- Our replication cohort sample sizes are much smaller than that of the discovery cohort. To discover 116 genome-wide associations, we used the data of 306,629 UKB participants; while to replicate these association findings, we used the data of 166,612 individuals in the BBJ, successfully re-identifying 41 associations -- still significant at an FDR of 0.05. We also tried to use the BioVU cohort, of which the sample size is 16,545; one-tenth of the BBJ's population, and thus, only found 15 associations still significant at an FDR of 0.05. We think a small sample size can partially account for the decline in the replication rate we observed here.
- In one of the cohorts, some loci were not genotyped or imputed for individuals: For example, in the BBJ data, 15 loci, involving 16 markers and associated signals, fell into this category; they were therefore not eligible for replication analysis in the first place, leaving 100 discovered associations to be subjected to the replication attempts. We have added this clarification to the Results and Methods 7.
- The cohorts we studied differ in their genetic ancestry: the BBJ cohort includes individuals of East Asian ancestry exclusively, while UKB and BioVU cohorts are predominantly Caucasian.

Please note that, in addition to filtering associations through the FDR test, we also compared association effect direction (Effect direction is a sign of the regression parameter measuring the putative phenotypic contribution of a given SNP in three cohorts). Effect directions in both the BBJ and BioVU cohorts are largely consistent with those discovered in the UKB cohort. In the BBJ cohort, 87 out of 100 associations were in the same effect direction. (Technically, there are 116 UKB-discovered associations, but 16 of these associations involved SNPs that were not genotyped or imputed in the BBJ.) In the BioVU cohort, 87 of 116 associations were in the same effect direction as in the UKB cohort. Moreover, 67 associations had the same effect directions in all three cohorts.

As suggested by the reviewer, we performed the additional analyses, reported below.

(1) We used the raw (unadjusted) p -values of 116 genetic associations identified in the discovery cohort to evaluate the excess (enrichment) of small p -values in the replication cohorts. The assumption underlying this analysis is this: If the 116 associations were, indeed, false positives, the analysis of the replication cohort's same marker associations would produce, on average, a random "replication" of about five percent out of the 116 associations, given the significance threshold of five percent.

Using the BBJ replication cohort, we found that 46 out of 100 discovered associations (46 percent) were replicated at the five percent significance level, which translates into a 9.2- fold enrichment of significance. With the BioVU replication cohort, we replicated 29 out of 116 discovered associations (25 percent) at the five percent significance level (which equals a five-fold enrichment of significance). Furthermore, we used both replication cohorts, the BBJ and the BioVU, jointly to identify 16 associations replicated at the five percent significance level across the two datasets. In this case, the apparent replication success rate equals to $16 / 108$ (14.81 percent; because 16 markers were unavailable in one of the replication datasets, 108 is the average of 100 and 116 associations). The probability of obtaining successful replications in two independent datasets simply by chance is five percent \times five percent, equalling 0.25 percent; therefore, we have a 59.2- fold enrichment of significance compared to that which is expected by chance. These calculations clearly suggest that the probability of our replication results being random false discovery matches is very low.

(2) We ran computer simulations to probe random "replication" frequency in cohorts derived from real genetic data and scrambled phenotypic data. To simulate

replication cohorts to match the size of the BBJ cohort, we used a repeated random sub-sampling of 166,612 individuals (the size of the BBJ cohort) out of the complete UKB Caucasian- ancestry cohort, including 306,629 unrelated individuals. In our analysis, a person- specific phenotype is represented by a vector of 20 real-valued numbers (the coordinates of an individual in the disease embedding space). A random permutation of the vector elements destroys true associations with a disease, generating simulated phenotypes devoid of any true genetic signal. Following this procedure, we generated a collection of replications cohorts subjected to the same analysis for the 116 loci as we applied to real replication cohorts (see Methods 7).

We applied GWAS methodology to the collection of synthetic replication cohorts, computing p -values for the significance of phenotypic data associations, with the set of 116 genetic markers identified in the discovery cohort and used the FDR method with a cut-off of 0.05 to account for multiple statistical tests. We repeated the generation of synthetic data, association analysis, and association validation 100 times. We discovered that, in 99 out of 100 experiments, none of the 116 associations were validated, while in one experiment, only one marker-phenotype association was significant. This analysis refutes the hypothesis that our replication results were predominantly random false associations.

(3) We compared our replication results with those in recent, GWAS-focused, peer- reviewed studies. Our replication statistics compare favorably to those published.

For example, Lopera-Maya *et al.* reported 37 associations between SNPs and the traits of bacterial taxa and pathways, and successfully replicated two significant associations in two independent cohorts (Lopera-Maya, et al., 2022, *Nature Genetics*). The COVID-19 host genetics initiative reported 13 genome-wide significant loci in association with SARS- CoV-2 infection or severe manifestations of COVID-19, and only suggested one locus that can be successfully replicated for its associations with severity or susceptibility (COVID-19 Host Genetics Initiative, 2021, *Nature*).

References:

Lopera-Maya, E.A., et al. Effect of host genetics on the gut microbiome in 7,738 participants of the Dutch Microbiome Project. *Nature Genetics* 2022;54(2):143-151.

COVID-19 Host Genetics Initiative, C.-H.G. Mapping the human genetic architecture of COVID-19. *Nature* 2021;600(7889):472-477.

3-Results: 'In order to computationally construct this disease space, we considered each patient's health history, chronologically ordered such that diseases experienced concurrent with one another (or soon before or after) share each other as context'. The hypothesis of chronological co-occurrence of diseases that are biologically linked is not necessarily what we observe in clinics. As an illustration, hypothyroidism can manifest twenty years after the onset of type 1 diabetes in certain patients, even though autoimmunity clearly drives the two disorders.

R1-3:

Agreed. We changed our prose to clarify this point. We also expanded the *Limitations* section of the manuscript to articulate this point. When complete medical histories of patients ("cradle-to-grave") are available, even long-term disease associations can be captured, at least in theory. In practice, we must work with shorter, partial medical histories, and some of the longer-term associations may appear lost.

4-Results: the authors may consider adding more details about the 20 dimensions that summarize 547 diseases. What did capture these 20 dimensions that may help us to understand the general mechanisms at work in disease physiopathology (e.g. brain development, inflammation, immunity, nutrition, oncogenesis...)?

R1-4:

Following the reviewer's suggestion, we included the orderings of diseases imposed by the remaining 19 dimensions in Extended Data Figure 2. To further facilitate disease space interpretation, we annotated each dimension by identifying a pair of disease categories that are most distant along the given dimension.

In our study, we used 547 unique diseases grouped into 21 disease categories; therefore, we additionally annotated space dimensions by the most distant disease categories for each dimension. For this purpose, we used the Wilcoxon rank sum test to determine the most separated disease categories for each dimension (see Methods 3 for technical details).

For example, Dimension 8 is labeled as "Ophthalmological (-) — Central Nervous System (+)," Dimension 2 as "Neuropsychiatric (-) — Infectious (+)," and Dimension 13

as “Infectious (-) — Cardiovascular (+)” (see Supplementary Fig. 2).

The disease embedding space encapsulates both diseases and patients, and thus, we can obtain intuition about human pathophysiology as follows:

- (1) The disease embedding space provides explicit distance/similarity between diseases (see Fig. 2a and 2b). We computed a data-driven nosology using our distance matrix for diseases (see Extended Data Fig. 1); the tree allows us to re-examine and even question the current nosology.

For example, migraine is a neighbor of eye inflammation in the ICD-9 cluster of “diseases of the central nervous system and sensory organs” (ICD-9 codes 320-389), according to the ICD-9 taxonomy (https://en.wikipedia.org/wiki/List_of_ICD-9_codes). Nevertheless, several recent studies suggested that migraine is closer to immune system diseases, such as irritable bowel syndrome (Gormley, et al., 2016, *Nature Genetics*; Wang, et al., 2017, *Nature Genetics*). Our disease embedding suggests that migraine is much closer to IBS with a similarity value of 0.48 than to eye inflammation, with a similarity value of 0.08. Furthermore, by comparing the 20-dimensional embedding vectors of the three diseases, we find that Dimension 8 appears to contribute most to similarity differences, since the largest positive values out of the 20-dimensional vectors for both of migraine and IBS, while most negative embedding value for eye inflammation, appear exactly on this dimension (see the vectors below, where their eight elements are highlighted in bold and are in underlined fonts).

Migraine = {0.842, -0.802, 0.0292, 0.852, -0.677, -0.0417, -0.110, **2.42**, 0.187, -0.188, -0.457, 0.462, 1.58, 0.641, 0.703, -1.66, -0.162, -1.02, -1.16, -0.00151},
IBS = {-0.520, -0.726, -2.47, 0.604, -0.145, 0.282, 1.21, **1.15**, 1.55, 1.08, -0.813, 0.625, 0.769, 0.912, 0.613, -0.543, -1.29, -0.608, -0.725, -0.00261},
Eye Inflammation = {1.13, -0.696, -1.69, -0.181, -0.677, 0.260, -1.01, **-1.55**, 0.499, -0.384, -0.249, 1.65, -0.107, -0.187, 0.524, -0.473, 0.0263, -0.517, -0.601, -0.0104}.

Next, we can suggest the disease physiopathology mechanisms by correlating this dimension with individuals’ genetics and many health-related phenotypes. Please see the individual-level discussion (2) below.

References:

Gormley, P., et al. Meta-analysis of 375,000 individuals identifies 38 susceptibility loci for migraine. *Nature Genetics* 2016;48(8):856-866.

Wang, K., et al. Classification of common human diseases derived from shared genetic and environmental determinants. *Nature Genetics* 2017;49(9):1319-+.

- (2) Functional annotations of genes in proximity to markers associated with disease space dimensions (116 strong associations) provide another glimpse at biological processes in human pathology. We used the NIH GWAS Catalog to identify known biological associations and then used gene-based pathway enrichment to annotate our findings further. In the revised version of the manuscript, we highlighted a few results from this analysis. For example, Dimension 8 associates with six genomic loci harboring genes: *BACH2*, *GTF3AP1*, *STAT6*, *EMSY*, and *SMAD3*. These are transcription factors regulating, in addition to other known functions, adaptive immune response by T- and B-lymphocytes, and synthesis of interleukin 4 and immunoglobulin E. This observation indirectly supports the conjecture of the immune etiology of such diseases as migraine and IBS. Another function attributed to these genes is that of regulating numbers of circulating eosinophils and neutrophils. These regulatory processes are implicated in asthma, hay fever, and other allergic diseases (see Supplementary Table 3 for the enrichment results specific to each of the 20 dimensions).

Further, we examined the associations between individual-specific disease embedding coordinates and a collection of 140 health-related phenotypes documented from the UKB cohort. These phenotypes measure ten general categories related to health, including blood count, blood biochemistry, urine biochemistry, spirometry, early life factors, anthropometry, addictions, diet, physical activity, and local environment (e.g., the coverage of green space/natural environment and air quality measured by nitrogen dioxide, nitrogen oxide, and particulate matter). For example, in Dimension 8, we found that measured counts of eosinophils and leukocytes are in strong negative associations with individual-specific embedding values in Dimension 8 (see Supplementary Fig. 4). Similarly, considering laboratory blood test values of creatinine and glycated hemoglobin HbA1c (see Supplementary Fig. 5) and micro-albumin in urine (see Supplementary Fig. 6), we found that these values are also strongly, negatively associated with Dimension 8 (the lower these measurements read for an individual, the greater the likelihood was that the individual would carry diseases like IBS or migraine; while the higher they read, the greater was the chance of the patient having eye inflammation). We have added this paragraph to the discussion session.

We described the methodology of these analyses in the “*Health-related Phenotypic Association Analyses of Embedding Dimensions*” section in Methods 9. The results are summarized in Supplementary Figs. 4–13.

Please also see R2-0.

5-Results: as the authors did not describe the biological mechanisms underlying the 20 dimensions, it is difficult to make sense of the 116 GWAS associations identified for these 20 outcomes.

R1-5:

Please see our reply to R1-4 above; in it, we try: (1) to annotate each of the 20 dimensions by the disease category pair that can be separated most widely in the respective dimensions (see Supplementary Fig. 2 and Extended Data Table 1); (2) to examine the disease clustering patterns in each of the 20 dimension, and; (3) to associate individuals’ embedding values with genetic and phenotypic information.

In addition, we add the following analysis results in order to make further sense of the 116 GWAS associations:

- (1) Out of our identified 116 associations that involve 108 unique loci, we try to pull the GWAS Catalog results for 68 loci reported by previous studies (see the Extended Data Table 3 and Supplementary Table 3). By referencing their studied phenotypes and linking them with our annotations about the 20 dimensions, we can interpret the biological mechanisms underlying the genetic association results.
- (2) We performed a pathway enrichment analysis (see the newly added Methods 8) and listed the enriched biological pathways or processes in Supplementary Table 3, confirming that these loci are associated with regulating adaptive immune response and interleukin 4 production. Immunity, such as interleukin 1 family signaling, is represented in Dimensions 1 and 10. Dimension 1 separates infectious and developmental diseases as two extremes, while Dimension 10 separates infectious and hematologic diseases. It is likely that the immune system is involved to some extent in every complex disease, and these two dimensions capture the interactions between weakened or over-active immune systems with other biological systems (see Supplementary Table 3 for the enrichment results specific to each of the 20 dimensions, and Methods 8 for

technical details). We have added this snippet to the results section.

Please also see R2-0.

6-Results: using the Japan Biobank (East Asian ancestry) to replicate GWAS associations from the UK Biobank (European ancestry) is highly questionable. Ethnic transferability of true GWAS association signals has been shown to be only partial in the literature.

R1-6:

We agree that the difference in ethnic/racial ancestry across discovery and validation cohorts is a legitimate concern.

Recent studies have demonstrated that GWAS results across different ethnic cohorts are, at least, partially transferable; see the paper by Li and Keating (2014, *Genome Medicine*) devoted to this issue. The following are examples of recent trans-ethnic GWAS studies listed by phenotype: lipoprotein levels (Coram, et al., 2013, *Am J Hum Genet*), blood pressure levels (Franceschini, et al., 2013, *Am J Hum Genet*), ischemic stroke and coronary artery disease (Dichgans, et al., 2014, *Stroke*), rheumatoid arthritis (Okada, et al., 2014, *Nature*), asthma (Lasky-Su, et al., 2012, *Clin Exp Allergy*), breast cancer (Siddiq, et al., 2012, *Hum Mol Genet*), and prostate cancer (Kote-Jarai, et al., 2011, *Nat Genet*).

Fortunately for our study, there is encouraging prior research demonstrating that GWAS for European and East Asian cohorts replicates reasonably well. For example, Lam et al. (*Nat. Genet* 2019) estimated that, in schizophrenia GWAS, the genetic correlation between genetic effect sizes for Europeans and East Asians reached 0.98 ± 0.03 . This group was able to run a meta-analysis by combining the two cohorts. Another study by Ntzani et al. (*Hum Genet* 2012) systematically assessed the consistency of 108 genome-wide associations across major ancestral groups (900 different cohorts). The estimated effect size correlation value for the European–East Asian comparison was 0.33, with a p -value of < 0.001 , which is much larger and more significant than comparisons for other ethnic groups. For example, corresponding values for European-African and Asian-African comparisons were 0.27 ($p = 0.20$) and 0.20 ($p = 0.51$), respectively.

The revised manuscript addresses the reviewer’s concern directly in Methods 7:

“We acknowledge that ethnicity differences between BBJ participants (East Asian ancestry) and the UKB participants for our GWAS discovery analyses (British

White ancestry) could possibly confound the genetic association tests, since allele frequencies and linkage disequilibrium patterns of the entire genome are not all similar across different ethnic populations (Peterson, et al., 2019, Cell). A slightly relieving fact is, specifically relevant to our study, historical research has shown that the overall correlations between the genetic association results concerning hundreds of different phenotypes based on European ethnic groups and those based on East Asian groups are very significant (Ntzani, et al., 2012, Hum Genet).”

References:

Coram, M.A., et al. Genome-wide characterization of shared and distinct genetic components that influence blood lipid levels in ethnically diverse human populations. **Am J Hum Genet** 2013;92(6):904-916.

Dichgans, M., et al. Shared genetic susceptibility to ischemic stroke and coronary artery disease: a genome-wide analysis of common variants. **Stroke** 2014;45(1):24-36.

Franceschini, N., et al. Genome-wide association analysis of blood-pressure traits in African-ancestry individuals reveals common associated genes in African and non- African populations. **Am J Hum Genet** 2013;93(3):545-554.

Kote-Jarai, Z., et al. Seven prostate cancer susceptibility loci identified by a multi-stage genome-wide association study. **Nat Genet** 2011;43(8):785-791.

Lasky-Su, J., et al. HLA-DQ strikes again: genome-wide association study further confirms HLA-DQ in the diagnosis of asthma among adults. **Clin Exp Allergy** 2012;42(12):1724-1733.

Li, Y.R. and Keating, B.J. Trans-ethnic genome-wide association studies: advantages and challenges of mapping in diverse populations. **Genome Med** 2014;6(10):91.

Lam, M., et al. Comparative genetic architectures of schizophrenia in East Asian and European populations. **Nat Genet** 2019;51(12):1670-1678.

Ntzani, E.E., et al. Consistency of genome-wide associations across major ancestral groups. **Hum Genet** 2012;131(7):1057-1071.

Okada, Y., et al. Genetics of rheumatoid arthritis contributes to biology and drug

discovery. *Nature* 2014;506(7488):376-381.

Siddiq, A., et al. A meta-analysis of genome-wide association studies of breast cancer identifies two novel susceptibility loci at 6q14 and 20q11. *Hum Mol Genet* 2012;21(24):5373-5384.

Peterson, R.E., et al. Genome-wide Association Studies in Ancestrally Diverse Populations: Opportunities, Methods, Pitfalls, and Recommendations. *Cell* 2019;179(3):589-603.

7-Results: estimating the predictive utility of the 116 GWAS association signals in only 5 diseases seems quite arbitrary, as the initial analysis included 547 diseases. We cannot exclude that the authors picked-up diseases that work the best. Summarizing the prediction ability for all diseases may make more sense.

R1-7:

We appreciate this concern but assure the Review and Editor that we did not cherry-pick diseases to test. Our choice of diseases was guided by two factors: (1) the value of disease prevalence translating into the largest case counts in the UK Biobank cohort; and (2) the availability of prior published disease-specific results for comparison with our own.

Following the reviewer's suggestion, we estimated and reported the performance of our models for all diseases with at least 15,000 case counts (considering the need to subsample this set of cases 100 times) in the UK Biobank cohort (see Supplementary Table 5). Our results turned out to be robust: The disease-embedding-based models consistently demonstrated a gain in the explanatory power of genetic variation (*i.e.*, greater Nagelkerke R^2). The accuracy of discriminating cases from controls by our models was greater than 60 percent for 12 of the 30 diseases we tested.

Reviewer #2 (Remarks to the Author):

The authors present an interesting study of the "human disease space" by organising co- occurring diseases in shared constellations. They identify some potentially valuable genetic associations with disease constellations and use genetic associations to define polygenic risk prediction.

I don't think the disease networks and resulting constellations are particularly novel,

although the approach is valid. However, use their use as a quantitative trait for GWAS may be novel and this is the most interesting aspect of manuscript. It would be interesting to understand the relationship between the GWAS associations to constellations and the individual constituent diseases. Particularly interesting are strong GWAS associations in loci that are not associated with constituent diseases. These may be the 44 loci described? What do these loci represent, what processes, and how might they impact disease development and multi-morbidity? This is not addressed in the manuscript, but seems like an important question?

R2-0:

Agreed. Please see R1-4, where we address, in detail, the annotation of the 20 embedding dimensions and their GWAS associations.

In line with the reviewer's suggestion, we made efforts in the following two aspects:

First, in order to interpret the identified GWAS associations, we:

- a. Searched the GWAS Catalog for our identified loci and their associated phenotypes as reported previously. When searching the GWAS Catalog for a match, we did not limit ourselves to constituent diseases but rather looked at any broad-sense phenotypes that had ever been reported (see the new Extended Data Table 3 and Supplementary Table 3), and;
- b. Conducted a pathway enrichment analysis based on GWAS summary statistics of individuals' embedding dimensions (see the new Supplementary Table 3).

Second, in order to improve our understanding of the disease embedding dimensions with which these loci associate, we now add greater details by:

- a. Annotating embedding dimensions by the most separable disease categories (see Methods 3), and;
- b. Performing health-related, phenotypic association analyses of embedding dimensions (see Methods 9).

By integrating the results from the two above aspects, we can now suggest what and how biological processes may impact multimorbidity and pathogenesis (please see R1-5 for details).

Detailed comments on the manuscript:

1) Abstract

The abstract does not cover the material very clearly and some of the claims for the methods are a little overstated, e.g. “We show that our disease-embedding-model manifests ubiquitous power in explaining phenotypic variation from genetic variation than conventional polygenic risk score models.” This is a bold statement which isn’t fully justified by the results. What is ubiquitous power for example?

R2-1-1:

We edited the abstract for clarity and toned down overly strong statements. The “ubiquitous power” was intended to suggest that our disease-embedding model encapsulates full nosology. Our disease-embedding model can explain the phenotypic variations of a great number of diseases from the exact same set of the discovered 116 genetic associations.

This is compared to the conventional polygenic risk score models, which are built on disease-specific genetic associations — meaning that different diseases would have to be paired with different sets of loci that were identified in disease-specific GWASs.

Please also see R1-7 for details regarding expanded benchmarking of the model’s predictive power towards many more diseases.

Sometimes the language is a little colloquial, for example, it’s not clear what the authors mean by terms such as distilled in this context.

R2-1-2:

Agreed. We made the language more formal throughout the manuscript. For example, in the sentence “we use ten dense disease constellations distilled from the space”, “distilled from the space” is now revised to “resulting from the clustering analysis of diseases in the embedding space”.

The abstract reports 44 novel associations, but these are not described in the manuscript!

R2-1-3:

The claim about 44 novel associations was based on the stable release of GWAS Catalog results dated 2019-07-30. We adjusted the number of novel associations from 44 to 40, based on a more recent version of the GWAS Catalog (dated 2022-05-17). Please see R2- 7-2 for the criteria we used to define novelty.

We marked the 40 novel loci with # symbols in Supplementary Table 2, reported analysis of the GWAS Catalog in Extended Data Table 3, and reported annotated enriched canonical pathways/biological processes in Supplementary Table 3. Please also see R2-0.

2) Introduction

The introduction refers briefly to the disease network concepts that are fundamental to this work, before moving on to focus on genetic theory. The objective to “transcend the arbitrariness of past schemes” is a reasonable one, but many arbitrary assumptions are still made, mainly about the role of genetics. Most notably, disease space underpinned by genetic space overlooks environment effects. How are demographics, environment, lifestyle and social deprivation accounted in this model. Could they not be included as this data is available in datasets like UK-Biobank.

R2-2-1:

We agree with the reviewer and strive to relax the earlier models’ strong assumptions, creating a tractable model. Nevertheless, a great deal more can be done in the future. It is usually unwise to start with the most complex approach before testing an ascending- complexity of simpler approaches. We were concerned that putting too much analysis in a single manuscript is bound to confuse future readers and diminish the utility of the manuscript.

Indeed, socio-demographic, and even environmental data, can be explicitly incorporated into our analysis, and we intend to do this in a future study that would then use the current study as baseline. We now add prose explicitly articulating this point to the Discussion section.

Finally, following the reviewer’s suggestion, we additionally tried to find various phenotypes’ associations with individual-specific disease embedding coordinates. In detail, we have tested a collection of 140 phenotypes that measured ten general

categories related to health, including blood count, blood biochemistry, urine biochemistry, spirometry, early life factors, anthropometry, addictions, diet, physical activity, and local environment. Particularly, the local environment indexes include the coverage of greenspace/natural environment and air quality measured by nitrogen dioxide, nitrogen oxide, and particulate matter. Admittedly, our analyses were restricted to the environmental measures recorded in the UK Biobank, and there are at least two limitations that we can think of:

(1) Many other undocumented environmental factors are likely to play more important roles, and;

(2) What we reported are asthma's correlations with the local environment, and it takes further functional experiments and analyses to infer causality.

Still, we hope our environmental analysis will generate some hypotheses to stimulate discussions and conjectures that researchers can test elsewhere.

Overall, the Introduction is too short and fails to introduce most of the concepts applied in this manuscript.

R2-2-2:

Agreed. We have now expanded the manuscript to better cover: (1) prior studies (see below and also R3-10); (2) methodological information regarding the word embedding algorithm, see R2-3-1 below, and; (3) future directions and limitations.

In addressing the first point, we have provided context by introducing previous efforts by other researchers to infer disease-disease relationships or disease networks (please also see our response of R3-10).

“There are prior studies aiming to infer disease-disease relationships based on genetic overlaps and comorbidity (Rzhetsky, et al., 2007, PNAS), protein-protein interactions (Goh, et al., 2007, PNAS), shared metabolism (Lee, et al., 2008, PNAS), and multiple types of input data at the simultaneously (Menche, et al., 2015, Science). The increasing availability of large-scale electronic health records (EHRs) has enabled researchers to identify disease-disease relationships in large scales. There is a set of studies using topological methods for disease analysis, focusing on one or a few specific diseases (Li, et al., 2015, Sci Transl Med); these analyses used EHRs for topological inference, sometimes incorporating the temporal order of diseases in a patient’s history, producing topological disease networks (Dagliati, et al., 2020, Artif Intell Med). Yet another group of studies generates disease networks

by computing pair-wise disease-disease correlations or relative risk scores (Hidalgo, et al., 2009, PLoS Comput Biol; Jensen, et al., 2014, Nat Commun)."

3) Results

The description of autoencoders would be better placed in the introduction.

R2-3-1:

Agreed, and we now add the following to the introduction (underlined fonts):

"In this study, we set out to develop and test a representation of the complete disorder spectrum as points in a continuous, metric, high-dimensional disease space, which then we linked to the genetic space that lies beneath it to enable clinical prediction and etiological discovery. Word embedding procedure was invented to capture semantic similarities of words and phrases in a natural language. It transforms one-hot representation of millions of words into a real-valued, low-dimensional space. The typical dimensionality of a word embedding is in the low hundreds. Each word is a point in this space, and semantically close words are located close to each other in the embedding space. In this study, we applied a word embedding technique to map diseases into a 20-dimensional embedding space, enabling new, downstream analyses. For example, we inferred disease classification in an objective, data-driven way (see Extended Data Fig. 1)."

The results section is rather brief and relies heavily on the figures. Little interpretation is performed in the results or discussion. More interpretation would improve the study.

R2-3-2:

Agree, and please also see R2-8 below.

We have made the following efforts to interpret our results and expand the results section:

- a. We've annotated the embedding dimensions by the most separable disease categories,
- b. Using Dimension 8 as an example, we listed the GWAS mapped genes and the enriched pathways.
- c. We searched the GWAS Catalog for our identified loci and their associated phenotypes that were reported previously.
- d. We've added details about our genetic correlation and heritability estimations.
- e. We've added the positive predictive values (PPV) and negative predictive values (NPV) for an additional set of 25 common diseases.

As for the discussion part, we:

- a. Expanded our interpretations of health-related phenotypic association analyses of the embedding dimensions, and;
- b. Suggested that superimposing non-genetic associations with diagnostic and genetic information for the disease space development shall be a desirable expansion in the future, and;
- c. Discussed our possible implication about genetic pleiotropy.

4) Figure 1

In the text, fig 1 is described as a projection of 20 dimensions based on PCA, however, the projection is T-SNE. No scale is indicated and more importantly T-SNE represents local structure only, and is inappropriate for representation of global structure which is the goal of this plot.

The application of principal component analysis is very poorly described in the methods and T-SNE is not described. Overall it was very hard to follow the process to generate the 20 dimensions. This seems like a fundamental weakness in the study and should be clarified throughout

R2-4: Accepted. We have revised our description to clarify that the 3-D projection shown in Fig. 1a was made using the *t*-SNE approach. We also thoroughly revised our method description to make our methodology clearer (see Methods 2). In a nutshell, we: (1) mapped diagnostic codes into 547 major disease groups; (2) computed 20-dimensional disease embedding using `word2vec` methodology, and; (3) used PCA analysis to uniquely rotate the embedding space, so that Dimension 1 corresponds to PC 1, Dimension 2 to PC 2, and so on. This revision forms an orthogonal coordinate system we eventually call our “disease embedding space” throughout this study.

5) Figure 2

I did not find figure 2 particularly informative, if used I suggest restricting to 2c only

R2-5:

As it is difficult to visualize 20-dimensional space properly, we used several lower-dimensional projections. Figures 2a and 2b show the closest disease pairs and the furthest pairs in the given space. We respectfully ask to retain the figure as it is, while explaining the figures better. Figures 2a and 2b show exemplar quantitative measures about disease proximity that our constructed disease embedding space can enable. This is worth highlighting in the form of figure presentation, especially considering that disease proximity, an abstract concept as it is, has traditionally been discussed

qualitatively but not quantitatively. Accordingly, we have improved the corresponding results session by adding more information.

6) Figure 3

Figure 3 is quite a good description of the phenotypic content of each constellation. Figure 3a would benefit from a better colour scheme. It is difficult to differentiate the colours, and the disease groupings are not always intuitive.

R2-6:

We made efforts to improve the color scheme by widening the color contrast and adding black-color frames to those disease groups that still cannot be clearly differentiated by color. As a result, we now separate disease groups by various high-contrast colors and, additionally, with or without black-color frames.

7) Figure 4

The figure is interesting and potentially highlights useful findings. It would be particularly interesting to explore pleiotropy. For example, there are 490 individual trait associations with HLA-B in the GWAS Catalog, how do these overlap with the constituents of the constellations that show association at the HLA-B locus? There is a lot of potential to explore pleiotropy and this might also highlight loci that show no or limited association to the traits in the constellations.

R2-7-1:

We agree completely. We would even argue that our study is mainly about pleiotropic connectivity across the entire collection of diseases. Our disease embedding space (mapped to genome dimension by dimension) encodes the shared etiology of multiple diseases.

We suggest that our study presents yet another type of pleiotropy; more than one disease space dimension associates with the same genetic marker. For example, the marker *rs34290285* (near *D2HGDH* locus) strongly associates with Dimensions 1 and 7; locus *HLA-B* (near markers *rs12212594*, *rs28380903*, *rs9265745*, *rs2428494*, *rs2523621*, and *rs2523616*) associates with Dimensions 1, 3, 4, 7, 12, and 16 (see Fig. 4c).

Further, Dimension 1 appears to influence a group of disorders, including autosomal abnormalities, chromosomal anomalies, and other developmental conditions. Dimension 3 is associated with predisposition to or protection against CNS infection, septicemia, and many other infectious diseases. Dimension 16 has an apparent effect on the absence/presence of diabetes mellitus, and other endocrine diseases.

Another way to examine pleiotropy is by looking at the proximity of diseases in the disease embedding space. For example, asthma's closest neighboring diseases are allergic rhinitis, emphysema COPD, chronic upper respiratory infection, chronic sinusitis, food allergy, alveolar disease, and pneumonia, which all belong to Constellation 3. In our disease embedding space, coordinates of asthma are {1.31, -0.452, -0.938, 0.197, -1.30, -0.426, 2.60, -1.26, 0.994, -1.15, -0.373, 0.969, 0.781, -0.295, -1.42, -0.897, 0.919, -0.350, -0.00992, 0.00486}. Dimensions 7, 15, 1, 5, and 8 have the largest absolute values, listed by decreasing value magnitude and highlighted in underlined, bold numerals. These five dimensions appear to represent the most salient properties of asthma-related disease collection; in our GWAS results, these five dimensions are associated with genetic loci (such as *HLA-B*) which are then significantly associated with asthma and its embedding space neighbors, such as allergic rhinitis, confirmed by matching these against the GWAS Catalog. These allergic disease constellation-linked loci often harbored the genes that encode proteins regulating immunity-related pathways, such as interleukin 1 receptor activity control, and interleukin 1 family signaling (see Supplementary Table 3). These associations align well with our current knowledge about the etiology of asthma and its constellation neighbors.

This two-pronged analysis is our blueprint for identifying pleiotropic genetic variation shared by constellations of etiologically similar diseases.

We also clarified the conceptual difference between constellations and dimension-specific genetic associations. Constellations are disease clusters in the 20-dimensional space (all 20 dimensions are used). Dimension-specific GWAS, as the term suggests, is done by treating a single dimension as a qualitative trait (only one of 20 coordinate values for each patient).

The abstract and text describing Fig 4 reports that 44 loci have never been reported in other association studies, yet these are not described or evaluated? What are these loci, how is their novelty determined? These are potentially very important but completely overlooked in the manuscript.

R2-7-2:

As described in R2-1-3, we have now adjusted the number of novel associations from 44 to 40, because we feel it is important to match our identified associations against

the most updated association results from the GWAS Catalog (dated 2022-05-17 for this revision).

We now mark the 40 novel loci with # symbols in Supplementary Table 2, report the details from the GWAS Catalog for the other loci that have been reported in previous studies in Extended Data Table 3, and report enriched canonical pathways/biological processes in Supplementary Table 3.

The novelty of our loci is determined via the following two steps:

Step 1: Download the most updated GWAS Catalog results (dated 2022-05-17).

Step 2: For each lead loci of the 108 loci that are involved in our identified associations, we try to search within its neighborhood ($\pm 5,000$ base pairs) for any loci in the catalog that have an r^2 value (a measure of linkage disequilibrium respective to the lead loci) greater than 0.1. If such loci cannot be found, then we would claim the novelty of the lead loci. We add this clarification in Methods.

Please also see R2-0 and R2-1-3.

8) Discussion

The discussion includes a reasonable consideration of the limitations of the study, but very limited depth in the consideration of the results. As discussed above, the study would benefit from more consideration of the meaning of the associations. Do they represent pleiotropy?

Do they represent common disease mechanisms? Do disease groupings in the constellations have common mechanisms between diseases that are otherwise dissimilar? The HLA association seems key to me here. Many diseases show HLA associations in GWAS indicating an autoimmune basis. Do the HLA associated constellations include autoimmune conditions? Overall the study raises many questions and would be strengthened by some consideration of mechanistic insights from the analysis.

R2-8:

We agree. Following the reviewer's suggestion, we have significantly expanded the discussion of our results.

A disease-disease closeness in the embedding space means that these diseases occur in similar contexts of other pathologies in the diagnostic histories of many patients. Therefore, this proximity may indicate shared disease etiology. The embedding space encodes a measure of similarity among diseases and thus allows us to re-examine the established nosology. For instance, in the International Disease Classification (version

9), migraine is grouped closely with eye inflammation in the cluster of “diseases of the central nervous system and sensory organs” (see ICD-9 codes 320-389). Alternatively, data-driven studies have suggested that migraine should be closer to immune system diseases, such as irritable bowel syndrome (IBS) (Gormley, et al., 2016, Nature Genetics; Wang, et al., 2017, Nature Genetics). Using the disease embedding space, we’ve estimated the cosine similarities between migraine and eye inflammation, and between migraine and IBS. Indeed, we found that migraine—IBS and migraine—eye inflammation similarity values were 0.48 and 0.08, respectively, suggesting migraine is, indeed, relatively closer to IBS. The analysis of the 20- dimensional vectors representing these diseases showed that Dimension 8 contributed most to the (dis)similarity among the three diseases. Along this dimension, migraine and IBS had the largest positive coordinate values compared to all the other coordinates of the two diseases. Eye inflammation’s position along this dimension, however, had the largest negative coordinate value (see our replies and the embedding vectors of Migraine, IBS, and Eye Inflammation in R1-4)

We now add the above extensions and qualifications to the discussion section. Next, we can speculate about the disease pathophysiology mechanisms behind it by correlating this dimension with individuals’ genetics and many health-related phenotypes. Please see the individual-level discussion (2) in R1-4, and in the current discussion section, where we introduce our efforts to associate individual-specific disease embedding coordinates with a collection of 140 various health-related phenotypes (also described in R2-2-2). These will also help to interpret the meaning of the embedding dimensions’ associations.

Please also see our response of R2-7-1 for a discussion on pleiotropy, which we will also add to the new discussion section.

Reviewer #3 (Remarks to the Author):

Authors present an interesting approach for multiple disease presentation and test how its mapping with genetic loci. The title states this approach is able to predict disease trajectories and risk, however is not clear what authors mean by “trajectories” and their definition.

Furthermore, prediction model performances are not good enough to state that the model is able to predict disease risks. I would change the title accordingly to these comments, underling other more important and innovative aspects of the proposed approach, such as the disease space representation.

R3-0-1:

We put effort into making the revised manuscript as clear as possible. We also acknowledge that, while our results for disease forecast from genetic data present an improvement over the state-of-the-art, prediction is still imprecise.

Time is not explicitly represented in the disease embedding space. However, each patient at every point of her life can be assigned to a unique 20-dimensional point in the disease space. Imagine that we have the whole life of a patient partitioned into, say, one-year segments. The patient's disease states for each time segment then represent a "walk" through the disease space. The chronologically ordered sequence of such states represents their health trajectory. Because in the US dataset, we can observe each patient for a limited number of years, this study's disease trajectory is compressed into a weighted average of health states.

To avoid further confusion the revised manuscript no longer mentions the term "trajectory." Following the reviewer's and the first reviewer's suggestions (see R1-1), we have changed the title of our manuscript to 'The High-dimensional Space of Human Diseases Built from Diagnosis Records and Mapped to Genetic Loci'.

In general, also given words restriction, the paper could be better structured in order to facilitate the understanding of some fundamental steps. Figures quality is in sometimes very poor and impede a proper interpretation of the results.

R3-0-2:

Agreed. We incorporated these suggestions by: (1) Expanding the introduction (see R2-2-2), and; (2) Put efforts into the interpretation of disease dimensions and GWAS results (see R1- 4). Also as suggested, we've included high-resolution figures in the revised manuscript.

Specific comments.

1. Line 103, the definition of trajectories, especially their sequential dimension, should be clearly introduced. This should be consistent across the whole paper. Note that the word "trajectories" is then only used here in line 134, but no mention to them is given in methods or results. At the end of the paper, the reader still doesn't have a clear idea of what a disease trajectory is.

R3-1:

Please also see our replies in R3-0-1.

2. Line 113. Which are the disease space dimension? Are they only spatial or also temporal?

R3-2:

Time is not explicitly represented in the disease embedding space. It is worth noting that the input to disease embedding development involved temporally ordered diseases, and therefore, neighborhood disease information was considered. Please see R3-0-1.

3. Line 115. What do you mean by “focal disease”?

R3-3:

In the chronological ordering of patient-specific diagnostic codes, the `word2vec` neural network is trained by masking one disease code in the sequence (“focal disease”) and then tries to predict this missing focal disease from the diseases immediately preceding and following the focal disease chronologically (“context diseases”). We included this explanation in the revised manuscript.

4. Line 125. While is clear that each patient is represented by a sequence of disease, is not clear how (or if) the temporal dimension has been considered.

R3-4:

With the disease embedding space constructed, we can represent 547 unique diseases and millions of patients as embedding vectors or space coordinates, and subsequently position them as points in the space. First, for disease-specific representation, there was no temporal dimension in the disease embedding output. In other words, we did not explicitly take into account the exact time when a diagnosis was identified, although the modeling took chronologically ordered disease records as input and considered disease neighborhood and sequence information through sliding windows of context (co-occurring) diseases. Second, for patient-specific representation, each patient’s coordinates in the space were defined as weighted mean disease embedding scores (see Equation 2 in Methods 5) over all the diseases contained in the patient’s diagnosis record. Please also see R3-0-1.

5. Line 127. Which is the strategy in defying the broader defined disease (i.e.

phenotypes) using ICD taxonomies? Have you considered approaches such as clinical classification system or PheCodes? Which could better include disease clinical/physiopathological information.

R3-5:

Although, indeed, we could have used PheCodes in our study (and we do use it in other studies which require a higher disease taxonomy resolution), for this study, we chose to use a more compact mapping for easier interpretation and to better handle the larger samples assigned to each mapped disease towards prediction model construction. To develop this more compact mapping, we started with a partial “bag of ICD-codes to disease” mapping initially provided to us by researchers at the Mayo Clinic. We further expanded the mapping with higher-level nodes from the International Classification of Diseases, with the addition of all Mendelian diseases that can be identified at the level of ICD codes. We used this mapping and made it available in several of our peer-reviewed publications:

*Rzhetsky, A., Wajngurt, D., Park, N., Zheng, T. Probing genetic overlap among complex human phenotypes. **Proc Natl Acad Sci U S A**, doi:10.1073/pnas.0704820104. (2007), PMC1906727, <https://www.ncbi.nlm.nih.gov/pmc/articles/PMC1906727/>*

*Blair, D. R., Lyttle, C. S., Mortensen, J. M., Bearden, C. F., Jensen, A. B., Khiabani, H., Melamed, R., Rabadan, R., Bernstam, E. V., Brunak, S., Jensen, L. J., Nicolae, D., Shah, N. H., Grossman, R. L., Cox, N. J., White, K. P., Rzhetsky, A. A nondegenerate code of deleterious variants in Mendelian loci contributes to complex disease risk. **Cell**, doi: 10.1016/j.cell.2013.08.030. (2013), PMC3844554, <https://www.ncbi.nlm.nih.gov/pmc/articles/PMC3844554/>*

*Wang, K., Gaitsch, H., Poon, H., Cox, N. J. & Rzhetsky, A. Classification of common human diseases derived from shared genetic and environmental determinants. **Nat Genet**, doi:10.1038/ng.3931 (2017), <https://www.ncbi.nlm.nih.gov/pubmed/28783162>*

*Jia, G. et al. Estimating heritability and genetic correlations from large health datasets in the absence of genetic data. **Nat Commun** 10, 5508, doi:10.1038/s41467-019-13455-0 (2019).PMC6890770, <https://www.ncbi.nlm.nih.gov/pubmed/31796735>*

6. While limitations regarding the choice of 20 dimension are discussed later in the manuscript, at this point of the paper I would provide some additional (even

empirical evidence) for this choice.

In general, several of the parameters choices could have been greatly impacted the final results, and affected the disease space representation. It would be important to compare different setting, possibly with a grid search approach.

R3-6:

Following the reviewer's suggestion, we experimented with varying the number of embedding dimensions. Specifically, we repeated embedding computations with 50 and 300 dimensions. We next computed the distances between all disease pairs (we implemented both cosine and Euclidian distance) within each embedding. Finally, we tested the correlation and linear regression correspondence between same-pair-of-diseases distances in comparisons of two different embeddings.

In our comparison of 20- against 50-dimensional embeddings: Pearson's correlation values were 0.74 when we used Euclidian distance (or 0.57 if we used cosine similarity), Student's t -test p -value $< 2.2 \times 10^{-16}$. In the corresponding linear regression analysis of the distances defined in these two embeddings, both the regression coefficient and intercept were significantly different from zero with p -value $< 2.2 \times 10^{-16}$ (see Supplementary Fig. 3 a and c for details).

Similarly, in comparing 50- and 300-dimensional embeddings: Pearson's correlation value was 0.62 if we used Euclidian distance (or 0.59 if we used cosine similarity), Student's t -test $p < 2.2 \times 10^{-16}$; and linear regression parameters were significantly different from zero with $p < 2.2 \times 10^{-16}$ (see Supplementary Fig. 3 b and d). It appears that dimension size variation produces robust results, but the choice of a higher embedding dimensionality (50 and 300) is harder to defend in practical applications with 547 distinct diseases in the embedding.

We added these results to Methods 2, Disease Embeddings.

7. Line 139. Please provide a more detailed definition of the point cloud.

R3-7:

After thorough consideration, we decided that the *point cloud* term is not necessary; we now reword the manuscript to avoid unnecessary definitions.

8. Line 142/143. I think this refers to the t -SNE projections, as indicated in figure 1 caption. It would be better to report this here too.

R3-8:

Yes, it referred to the *t*-SNE projection, and we have added the following clarification:

“Because it is difficult to visualize an object in a 20-dimensional disease space directly, we proposed multiple projections of it to facilitate an understanding of its properties. As one type of projection, Plate a in Fig. 1 shows a three-dimensional projection of the disease space through the t-Distributed Stochastic Neighbor Embedding (t-SNE) algorithm, with an emphasis on neuropsychiatric diseases (three shades of red), infections (green), and cancers (yellow).”

9. “Our disease embedding space contradicts some intuitions formed from living in two and three dimensions.” This sentence is not clear.

R3-9:

In light of the reviewer’s comment, we removed this vague sentence, and now explain our original intention more clearly by suggesting that we represented disease proximity in terms of their angles (*i.e.*, cosine similarity) between vectors in order to glimpse the properties of our disease embedding space.

10. Disease proximity is represented via cosine similarity. Your approach can be compared to topological data analyses approaches (see “Identification of type 2 diabetes subgroups through topological analysis of patient similarity” by Li), not only for this specific choice but also because its ability to represent of disorder spectrum as points in a continuous space, which is foremost important in the context of precision medicine. I would refer to these previous attempts and results, even if applied in specific diseases contexts. Some topological data analyses applications (see “Using topological data analysis and pseudo time series to infer temporal phenotypes from electronic health records” by Dagliati) can also deal with disease trajectories in time.

R3-10:

Agreed. We include the proposed references and now refer to them in the introductory section.

References:

Dagliati, A., et al. Using topological data analysis and pseudo time series to infer temporal phenotypes from electronic health records. Artif Intell Med 2020;108:101930.

Goh, K.I., et al. The human disease network. Proc Natl Acad Sci U S A 2007;104(21):8685-8690.

Jensen, A.B., et al. Temporal disease trajectories condensed from population-wide registry data covering 6.2 million patients. *Nat Commun* 2014;5:4022.

Lee, D.S., et al. The implications of human metabolic network topology for disease comorbidity. *Proc Natl Acad Sci U S A* 2008;105(29):9880-9885.

Li, L., et al. Identification of type 2 diabetes subgroups through topological analysis of patient similarity. *Sci Transl Med* 2015;7(311):311ra174.

Menche, J., et al. Disease networks. Uncovering disease-disease relationships through the incomplete interactome. *Science* 2015;347(6224):1257601.

Zheng, C. and Xu, R. Large-scale mining disease comorbidity relationships from post-market drug adverse events surveillance data. *BMC Bioinformatics* 2018;19(Suppl 17):500.

11. Figure 2c. These are very interesting results, however it looks a bit of “cherry picking”, why have you chosen to show this dimension?

R3-11:

Apology for not showing all the dimensions, and instead randomly selecting just one for display. Now, we have provided analogous plots for all 20 dimensions and have reported them in Extended Data Fig. 2.

12. I would respectfully argue that SVD is a recently developed approach. Also, while k-SVD clustering might sound good if applied to your raw data, as you've already computed the embeddings there are other approaches that might work as well? Again, it might be worth to compare the topological approaches, where the clustering step is often performed via hierarchical clustering, or different clustering techniques can be compared.

R3-12:

We agree—alternative methods are indeed available. We decided to use *k*-SVD method, because it had a proven success of applying to word embedding methods particularly. This success was mathematically explained in the following two articles that were widely cited, *i.e.*, 270 and 180 citations, respectively:

Arora, S., Li, Y., Liang, Y., Ma, T. & Risteski, A. A latent variable model approach to PMI-based word embeddings. *Transaction of Association for Computational*

Linguistics, pages 385–399 (2016).

Arora, S., Li, Y., Liang, Y., Ma, T. & Risteski, A. *Linear algebraic structure of word senses, with applications to polysemy*. *arXiv 1601.03764* (2016).

Following the reviewer's suggestion, we implemented the suggested topological methods. We used a cosine similarity to measure a between-disease distance. We then computed average similarities between pairs of clusters (constellations, see Extended Data Table 2).

We then compared ten constellations produced by hierarchical clustering and by the original k -SVD method. We used a hypergeometric enrichment test to probe similarities of results produced by these two methods. We found that: (1) There were one-to-one correspondences between the two sets of constellations; (2) The resulting p -values measuring the probability to obtain this correspondence by chance were 3.2×10^{-9} , 2.5×10^{-34} , 1.1×10^{-3} , 1.9×10^{-25} , 2.8×10^{-7} , 6.4×10^{-21} , 2.1×10^{-27} , 9.5×10^{-16} , 1.7×10^{-15} , and 6.5×10^{-4} for the constellations from the first to tenth constellations, respectively, suggesting the clustering results are robust against different clustering methods.

We included the above descriptions in Methods 4.

13. Figure 3 (especially panel b) quality is poor, and it does not allow to read results.

R3-13:

Agreed; we fixed this

14. Lines 198 and line 203/204. Again, I would better clarify how the consequentially similar diseases are represented, that is disease trajectory. Line 203/204 might explain my previous point regarding what each object (spheres in figure 1) in the disease embedding space represents. It would be worth to better explain this in the previous section. An example might facilitate the reader to have a better idea of the disease space/trajectories.

R3-14:

Agreed. We answer these excellent questions concerning disease trajectories, temporal information, and the meaning of objects in the disease space above; please see our replies in R3-0-1.

15. Figure 3 is a bit confusing. It includes constellations (at this point not yet exploited for the estimates of genetic correlation) and dimensions with their genetic correlation estimations. I think it would be clearer to include 3b and 3c with figure 4.

R3-15:

Following the reviewer's suggestion, we now merge Figure 3b and Figure 3c with Figure 4 to create a new Figure 4. On the one hand, we have improved the figure legend for the newly merged Figure 4a and Figure 4b; on the other, we've added the following details about heritability estimations and genetic correlations to the Results section (please see the underlined content):

“Note that GWAS-based heritability estimates appear much smaller than family-based heritability, showing that gene-gene and gene-environment interactions may play a role in the genetics of these 20 dimensions, interpreted as traits — besides rare genetic variation. Our analysis suggests that among all dimensions, Dimension 19, “Immune (-) — Neuropsychiatric (+),” has both the largest family heritability and the parental e^2 . Dimension 15, “Metabolic (-) — Infectious (+),” has the lowest values of these family-based estimates. As shown in Fig. 4b, these 20 continuous traits possess complex patterns of pair-wise genetic correlations, which we estimated using both the US MarketScan’s family data (the upper triangular matrix) and the BBJ’s GWAS data (the lower triangular matrix). Estimates are distinct across the two approaches, but are significantly positively correlated ($r = 0.224$, with a 95 percent credible interval [0.085, 0.355], $p = 1.85 \times 10^{-3}$). While the individual heritability of a single dimension is estimated below 0.2, due to high genetic correlation between dimensions, the joint co-heritability of all 20 dimensions is 0.187. For example, the family- and the GWAS-based approaches agree that Dimension 9, “Integumentary (-) — Neuropsychiatric (+),” has significantly positive genetic correlations with Dimensions 6, 3, and 16, but has negative genetic correlations with Dimensions 13 and 2.”

16. Lines 309/310. Prediction accuracy is always very low, might be interesting to include PPV and NPV to better understand the reason for poor performance.

R3-16:

Agreed and done. PPV and NPV are similarly valued as is prediction accuracy. Please also see our responses in R1-7.

A key point about our modeling practice is that we relied on the universal set of the

discovered 116 genetic associations to build prediction models for a large number of various diseases, achieving better power in explaining their phenotypic variation and greater (or equivalent) accuracy in predicting disease liability, compared to conventional models built on disease-specific genetic associations. It is worth noting that complex disease liability prediction, purely based on genetics, is known to be a difficult problem, considering that many environmental factors could play important roles; in this regard, researchers typically pursue improved model explainability, owing to more relevant genetic loci found or advanced models constructed. In addition, our modeling endeavor here suggests that the disease embedding space is, indeed, biologically relevant, and its other potentials for deciphering inter-connections between diseases could open many future research and medical opportunities.

Following the reviewer's suggestion, we now add PPV and NPV as performance indexes for the five, representative common diseases (see Supplementary Table 4) and for all other diseases that have a number of cases more than 15,000, *i.e.*, twice of the sum of 5,000 cases for model training and 2,500 cases for model testing (there are 25 diseases in total; see Supplementary Table 5). We find that prediction accuracy, PPV, and NPV are similarly valued for each of the reported diseases, suggesting that the prediction performance was reliable and balanced. Out of the 30 diseases about which we now report, there are twelve diseases for which model prediction accuracies are greater than 60 percent. In addition, our embedding-vector-based models can provide Nagelkerke R^2 that are always greater than conventional models built on disease-specific GWAS Catalog loci, indicating that our models proposed unanimous gain in the explanatory power of genetic variation.

Decision Letter, first revision:

Date: 28th February 23 05:02:06

Last Sent: 28th February 23 05:02:06

Triggered By: Fernando Chirigati

From: fernando.chirigati@us.nature.com

To: arzhetsky@uchicago.edu

CC: computacionalscience@nature.com

BCC: fernando.chirigati@us.nature.com

Subject: AIP Decision on Manuscript NATCOMPUTSCI-21-0888A

Message: Our ref: NATCOMPUTSCI-21-0888A

28th February 2023

Dear Dr. Rzhetsky,

Thank you for submitting your revised manuscript "The High-dimensional Space of Human Diseases Built from Diagnosis Records and Mapped to Genetic Loci" (NATCOMPUTSCI-21-0888A). It has now been seen by the original Referee #3 and their comments are below. Unfortunately, Referees #1 and #2 were unresponsive, and Referee #3 provide comments on the other responses as well for us. This reviewer find that the paper has improved in revision, and therefore we'll be happy in principle to publish it in Nature Computational Science, pending minor revisions to satisfy the referees' final requests and to comply with our editorial and formatting guidelines.

We would like to apologize for the delay in reaching a decision about your paper, as we have been chasing referees for some time.

In the meantime, please complete a reporting summary that collects information on experimental design at your earliest convenience, sending it over via email by responding to this letter: <https://www.nature.com/documents/nr-reporting-summary.zip>

TRANSPARENT PEER REVIEW

Nature Computational Science offers a transparent peer review option for original research manuscripts. We encourage increased transparency in peer review by publishing the reviewer comments, author rebuttal letters and editorial decision letters if the authors agree. Such peer review material is made available as a supplementary peer review file. **Please state in the cover letter 'I wish to participate in transparent peer review' if you want to opt in, or 'I do not wish to participate in transparent peer review' if you don't.** Failure to state your preference will result in delays in accepting your manuscript for publication.

Please note: we allow redactions to authors' rebuttal and reviewer comments in the interest of confidentiality. If you are concerned about the release of confidential data, please let us know specifically what information you would like to have removed.

Please note that we cannot incorporate redactions for any other reasons. Reviewer names will be published in the peer review files if the reviewer signed the comments to authors, or if reviewers explicitly agree to release their name. For more information, please refer to our [FAQ page](https://www.nature.com/documents/nr-transparent-peer-review.pdf).

Thank you again for your interest in Nature Computational Science Please do not hesitate to contact me if you have any questions.

Best,

Fernando

--

Fernando Chirigati, PhD
Chief Editor, Nature Computational Science
Nature Portfolio

ORCID

Reviewer #3 (Remarks to the Author):

Thank you for the detailed responses provided. The additional analyses greatly improved the manuscript, such as the efforts to clarify several methodological steps and justify the choice of specific settings.

In order to avoid misunderstanding, in section 4 of Methods I would not state "topological methods, such as hierarchical clustering. "
As topological methods, while based on a step of hierarchical clustering, add further computations based on overlapping regions of the similarity projections.

Author Rebuttal, second revision:

Reviewer #3 (Remarks to the Author):

Thank you for the detailed responses provided. The additional analyses greatly improved the manuscript, such as the efforts to clarify several methodological steps and justify the choice of specific settings.

In order to avoid misunderstanding, in section 4 of Methods I would not state "topological methods, such as hierarchical clustering. "
As topological methods, while based on a step of hierarchical clustering, add further computations based on overlapping regions of the similarity projections.

Removed as suggested.

Final Decision Letter:**Date:** 13th April 23 14:43:00**Last Sent:** 13th April 23 14:43:00**Triggered By:** Fernando Chirigati**From:** fernando.chirigati@us.nature.com**To:** arzhetsky@uchicago.edu**BCC:** computacionalscience@nature.com,rjsproduction@springernature.com,rjsart@springernature.com,fernando.chirigati@us.nature.com**Subject:** Decision on Nature Computational Science manuscript NATCOMPUTSCI-21-0888B**Message** Dear Professor Rzhetsky,

:

We are pleased to inform you that your Article "The High-dimensional Space of Human Diseases Built from Diagnosis Records and Mapped to Genetic Loci" has now been accepted for publication in Nature Computational Science.

Once your manuscript is typeset, you will receive an email with a link to choose the appropriate publishing options for your paper and our Author Services team will be in touch regarding any additional information that may be required.

Please note that *Nature Computational Science* is a Transformative Journal (TJ). Authors may publish their research with us through the traditional subscription access route or make their paper immediately open access through payment of an article-processing charge (APC). Authors will not be required to make a final decision about access to their article until it has been accepted. [Find out more about Transformative Journals](https://www.springernature.com/gp/open-research/transformative-journals)

Authors may need to take specific actions to achieve [compliance with funder and institutional open access mandates](https://www.springernature.com/gp/open-research/funding/policy-compliance-faqs). If your research is supported by a funder that requires immediate open access (e.g. according to [Plan S principles](https://www.springernature.com/gp/open-research/plan-s-compliance)) then you should select the gold OA route, and we will direct you to the compliant route where possible. For authors selecting the subscription publication route, the journal's standard licensing terms will need to be accepted, including [self-archiving policies](https://www.springernature.com/gp/open-research/policies/journal-policies). Those licensing terms will supersede any other terms that the author or any third party may assert apply to any version of the manuscript.

If you have any questions about our publishing options, costs, Open Access requirements,

or our legal forms, please contact ASJournals@springernature.com

Acceptance of your manuscript is conditional on all authors' agreement with our publication policies (see <https://www.nature.com/natcomputsci/for-authors>). In particular your manuscript must not be published elsewhere and there must be no announcement of the work to any media outlet until the publication date (the day on which it is uploaded onto our web site).

Before your manuscript is typeset, we will edit the text to ensure it is intelligible to our wide readership and conforms to house style. We look particularly carefully at the titles of all papers to ensure that they are relatively brief and understandable.

Once your manuscript is typeset and you have completed the appropriate grant of rights, you will receive a link to your electronic proof via email with a request to make any corrections within 48 hours. If, when you receive your proof, you cannot meet this deadline, please inform us at rjsproduction@springernature.com immediately.

If you have queries at any point during the production process then please contact the production team at rjsproduction@springernature.com. Once your paper has been scheduled for online publication, the Nature press office will be in touch to confirm the details.

Content is published online weekly on Mondays and Thursdays, and the embargo is set at 16:00 London time (GMT)/11:00 am US Eastern time (EST) on the day of publication. If you need to know the exact publication date or when the news embargo will be lifted, please contact our press office after you have submitted your proof corrections. Now is the time to inform your Public Relations or Press Office about your paper, as they might be interested in promoting its publication. This will allow them time to prepare an accurate and satisfactory press release. Include your manuscript tracking number NATCOMPUTSCI-21-0888B and the name of the journal, which they will need when they contact our office.

About one week before your paper is published online, we shall be distributing a press release to news organizations worldwide, which may include details of your work. We are happy for your institution or funding agency to prepare its own press release, but it must mention the embargo date and Nature Computational Science. Our Press Office will contact you closer to the time of publication, but if you or your Press Office have any inquiries in the meantime, please contact press@nature.com.

We welcome the submission of potential cover material (including a short caption of around 40 words) related to your manuscript; suggestions should be sent to Nature Computational Science as electronic files (the image should be 300 dpi at 210 x 297 mm in either TIFF or JPEG format). We also welcome suggestions for the Hero Image, which appears at the top of our <http://www.nature.com/natcomputsci> home page; these should be 72 dpi at 1400 x 400 pixels in JPEG format. Please note that such pictures should be selected more for their aesthetic appeal than for their scientific

content, and that colour images work better than black and white or grayscale images. Please do not try to design a cover with the Nature Computational Science logo etc., and please do not submit composites of images related to your work. I am sure you will understand that we cannot make any promise as to whether any of your suggestions might be selected for the cover of the journal.

Best,
Fernando

--

Fernando Chirigati, PhD
Chief Editor, Nature Computational Science
Nature Portfolio

P.S. Click on the following link if you would like to recommend Nature Computational Science to your librarian: <https://www.springernature.com/gp/librarians/recommend-to-your-library>

** Visit the Springer Nature Editorial and Publishing website at <http://editorial-jobs.springernature.com> for more information about our career opportunities. If you have any questions please click [here](mailto:editorial.publishing.jobs@springernature.com).**